# FEW-SHOT TEXT CLASSIFICATION WITH DISTRIBUTIONAL SIGNATURES

**Yujia Bao[†*], Menghua Wu[†*], Shiyu Chang[‡], Regina Barzilay[†]**
[†]Computer Science and Artificial Intelligence Lab, MIT
[‡]MIT-IBM Watson AI LAB, IBM Research
{yujia,rmwu,regina}@csail.mit.edu, {shiyu.chang}@ibm.com

## ABSTRACT

In this paper, we explore meta-learning for few-shot text classification. Meta-learning has shown strong performance in computer vision, where low-level patterns are transferable across learning tasks. However, directly applying this approach to text is challenging–lexical features highly informative for one task may be insignificant for another. Thus, rather than learning solely from words, our model also leverages their distributional signatures, which encode pertinent word occurrence patterns. Our model is trained within a meta-learning framework to map these signatures into attention scores, which are then used to weight the lexical representations of words. We demonstrate that our model consistently outperforms prototypical networks learned on lexical knowledge (Snell et al., 2017) in both few-shot text classification and relation classification by a significant margin across six benchmark datasets (20.0% on average in 1-shot classification).[1]

## 1 INTRODUCTION

In computer vision, meta-learning has emerged as a promising methodology for learning in a low-resource regime. Specifically, the goal is to enable an algorithm to expand to new classes for which only a few training instances are available. These models learn to generalize in these low-resource conditions by recreating such training episodes from the data available. Even in the most extreme low-resource scenario–a single training example per class–this approach yields 99.6% accuracy on the character recognition task (Sung et al., 2018).

Given this strong empirical performance, we are interested in employing meta-learning frameworks in NLP. The challenge, however, is the degree of transferability of the underlying representation learned across different classes. In computer vision, low-level patterns (such as edges) and their corresponding representations can be shared across tasks. However, the situation is different for language data where most tasks operate at the lexical level. Words that are highly informative for one task may not be relevant for other tasks. Consider, for example, the corpus of HuffPost headlines, categorized into 41 classes. Figure 1 shows that words highly salient for one class do not play a significant role in classifying others. Not surprisingly, when meta-learning is applied directly on lexical inputs, its performance drops below a simple nearest neighbor classifier. The inability of a traditional meta-learner to zoom-in on important features is further illustrated in Figure 2: when considering the target class *fifty* (lifestyle for middle-aged), the standard prototypical network (Snell et al., 2017) attends to uninformative words like "date," while downplaying highly predictive words such as "grandma."

In this paper we demonstrate that despite these variations, we can effectively transfer representations across classes and thereby enable learning in a low-resource regime. Instead of directly considering words, our method utilizes their *distributional signatures*, characteristics of the underlying word distributions, which exhibit consistent behaviour across classification tasks. Within the meta-learning framework, these signatures enable us to transfer attention across tasks, which can consequently be used to weight the lexical representations of words. One broadly used example of such distributional

---

[*]Equal contribution.

[1]Our code is available at `https://github.com/YujiaBao/Distributional-Signatures`.

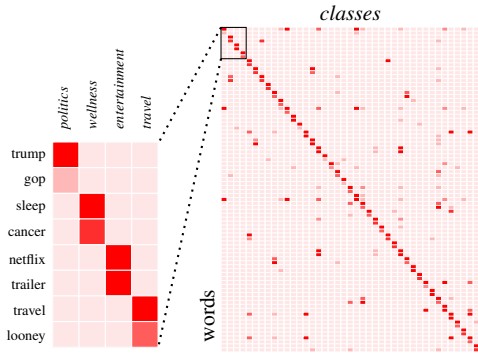

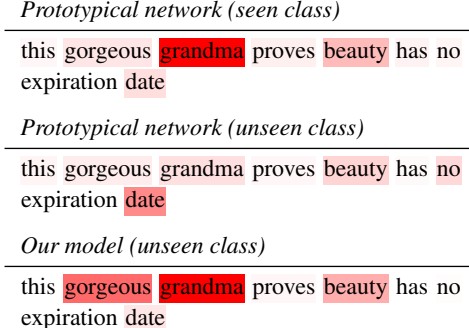

Figure 1: Different classes exhibit different word distributions in HuffPost headlines. We compute the local mutual information (LMI) (Evert, 2005) between words and classes. For each class, we include its top 2 LMI-ranked words. Darker colors indicate higher LMI.

Figure 2: Visualization of word importance on example from class *fifty* in HuffPost headlines. Top: *fifty* is seen during meta-training; prototypical network (Snell et al., 2017) finds important words. Middle: *fifty* is unavailable during meta-training; it fails to generalize. Bottom: Our model identifies key words for unseen classes.

signatures is tf-idf weighting, which explicitly specifies word importance in terms of its frequency in a document collection, and its skewness within a specific document.

Building on this idea, we would like to learn to utilize distributional signatures in the context of cross-class transfer. In addition to word frequency, we assess word importance with respect to a specific class. This latter relation cannot be reliably estimated of the target class due to the scarcity of labeled data. However, we can obtain a noisy estimate of this indicator by utilizing the few provided training examples for the target class, and then further refine this approximation within the meta-learning framework. We note that while the representational power of distributional signatures is weaker than that of their lexical counterparts, meta knowledge built on distributional signatures are better able to generalize.

Our model consists of two components. The first is an *attention generator*, which translates distributional signatures into attention scores that reflect word importance for classification. Informed by the attention generator's output, our second component, a *ridge regressor*, quickly learns to make predictions after seeing only a few training examples. The attention generator is shared across all episodes, while the ridge regressor is trained from scratch for each individual episode. The latter's prediction loss provides supervision for the attention generator. Theoretically, we show that the attention generator is robust to word-substitution perturbations.

We evaluate our model on five standard text classification datasets (Lang, 1995; Lewis et al., 2004; Lewis, 1997; He & McAuley, 2016; Misra, 2018) and one relation classification dataset (Han et al., 2018). Experimental results demonstrate that our model delivers significant performance gains over all baselines. For instance, our model outperforms prototypical networks by 20.6% on average in one-shot text classification and 17.3% in one-shot relation classification. In addition, both qualitative and quantitative analyses confirm that our model generates high-quality attention for unseen classes.

## 2 RELATED WORK

**Meta-learning** Meta-learning has been shown to be highly effective in computer vision, where low-level features are transferable across classes. Existing approaches include learning a metric space over input features (Koch, 2015; Vinyals et al., 2016; Snell et al., 2017; Sung et al., 2018), developing a prior over the optimization procedure (Ravi & Larochelle, 2016; Finn et al., 2017; Nichol & Schulman, 2018; Antoniou et al., 2018), and exploiting the relations between classes (Garcia & Bruna, 2017). These methods have been adapted with some success to specific applications in NLP, including machine translation (Gu et al., 2018), text classification (Yu et al., 2018; Geng et al., 2019; Guo et al., 2018; Jiang et al., 2019) and relation classification (Han et al., 2018). Such models primarily build meta-knowledge on lexical representations.

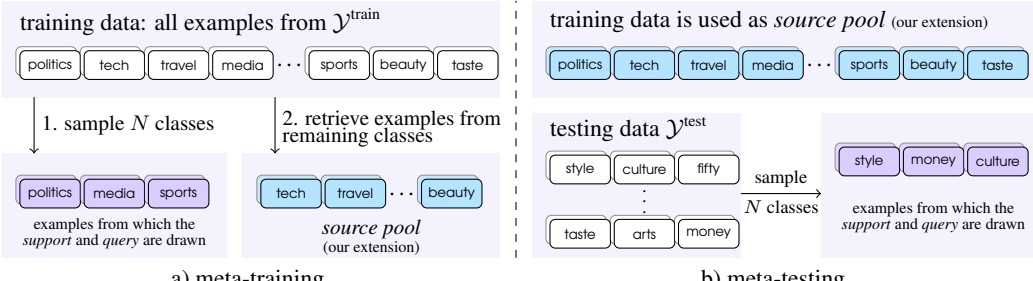

Figure 3: Episode generation. a) Meta-training: First, sample $N$ classes from $\mathcal{Y}^{\text{train}}$. Then, sample the support set and the query set from the $N$ classes. We use examples from the remaining classes to form the source pool. b) Meta-testing: Select $N$ new classes from $\mathcal{Y}^{\text{test}}$ and sample the support set and the query set from these $N$ classes. We use all examples from $\mathcal{Y}^{\text{train}}$ to form the source pool.

However, as our experiments show, there exist innate differences in transferable knowledge between image data and language data, and lexicon-aware meta-learners fail to generalize on standard multi-class classification datasets.

In this work, we observe that even though salient features in text may not be transferable, their *distributional* behaviors are alike. Thus, we focus on learning the connection between word importance and distributional signatures. As a result, our model can reliably identify important features from novel classes.

**Transfer learning** Our work is also closely related to transfer learning: we assume access to a large number of labeled examples from *source* classes, and we would like to identify word importance for the *target* classification task. Current approaches transfer knowledge from the source to the target by either fine-tuning a pre-trained encoder (Howard & Ruder, 2018; Peters et al., 2018; Radford et al., 2018; Bertinetto et al., 2019), or multi-task learning with a shared encoder (Collobert & Weston, 2008; Liu et al., 2015; Luong et al., 2015; Strubell et al., 2018). Recently, Bao et al. (2018) also successfully transferred task-specific attention through human rationales.

In contrast to these methods, where the transfer mechanism is pre-designed, we learn to transfer based on the performance of downstream tasks. Specifically, we utilize distributional statistics to transfer attention across tasks. We note that while Wei et al. (2017) and Sun et al. (2018) also learn transfer mechanisms for image recognition, their methods do not directly apply to NLP.

## 3 BACKGROUND

In this section, we first summarize the standard meta-learning framework for few-shot classification and describe the terminology (Vinyals et al., 2016). Next, we introduce our extensions to the framework. Figure 3 and 4 graphically illustrate our framework.

**Problem statement** Suppose we are given labeled examples from a set of classes $\mathcal{Y}^{\text{train}}$. Our goal is to develop a model that acquires knowledge from these training data, so that we can make predictions over new (but related) classes, for which we only have a few annotations. These new classes belong to a set of classes $\mathcal{Y}^{\text{test}}$, disjoint from $\mathcal{Y}^{\text{train}}$.

**Meta-training** In meta-learning, we emulate the above testing scenario during meta-training so our model learns to quickly learn from a few annotations. To create a single *training episode*, we first sample $N$ classes from $\mathcal{Y}^{\text{train}}$. For each of these $N$ classes, we sample $K$ examples as our training data and $L$ examples as our testing data. We update our model based on loss over these testing data. Figure 4a shows an example of an episode. We repeat this procedure to increase the number of training episodes, each of which is constructed over its own set of $N$ classes. In literature, the training data of one episode is commonly denoted as the *support set*, while the corresponding testing data is known as the *query set*. Given the support set, we refer to the task of making predictions over the query set as $N$-*way* $K$-*shot classification*.

**Meta-testing** After we have finished meta-training, we apply the same episode-based mechanism to test whether our model can indeed adapt quickly to new classes. To create a *testing episode*, we

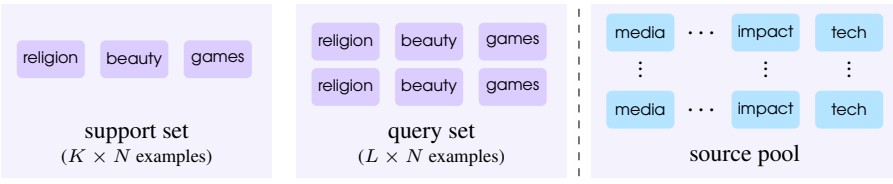

a) traditional episode          b) our extension

Figure 4: Single episode with $N = 3$, $K = 1$, $L = 2$. Rectangles denote input examples. The text inside corresponds to their labels. An episode contains a support set, a query set, and a source pool.

first sample $N$ new classes from $\mathcal{Y}^{\text{test}}$. Then we sample the support set and the query set from the $N$ classes. We evaluate the average performance on the query set across all testing episodes.

**Our extension**  We observe that even though all examples from $\mathcal{Y}^{\text{train}}$ are accessible throughout meta-training, the standard meta-learning framework (Vinyals et al., 2016) only learns from small subsets of these data per training episode. In contrast, our model leverages distributional statistics over all training examples for more robust inference. To accommodate this adjustment, we augment each episode with a *source pool* (Figure 4b). During meta-training (Figure 3a), this source pool includes all examples from training classes not selected for the particular episode. During meta-testing (Figure 3b), this source pool includes all training examples.

## 4  METHOD

**Overview**  Our goal is to improve few-shot classification performance by learning high-quality attention from the distributional signatures of the inputs. Given a particular episode, we extract relevant statistics from the source pool and the support set. Since these statistics only roughly approximate word importance for classification, we utilize an *attention generator* to translate them into high-quality attention that operates over words. This generated attention provides guidance for the downstream predictor, a *ridge regressor*, to quickly learn from a few labeled examples.[2]

We note that the attention generator is optimized over all training episodes, while the ridge regressor is trained from scratch for each episode. Figure 5 illustrates the two components of our model.

- **Attention generator:** This module generates class-specific attention by combining the distributional statistics of the source pool and the support set (Figure 5a). The generated attention provides the ridge regressor an inductive bias on the word importance. We train this module based on feedback from the ridge regressor (Section 4.1).

- **Ridge regressor:** For each episode, this module constructs lexical representations using the attention derived from distributional signatures (Figure 5b). The goal of this module is to make predictions over the query set, after learning from the support set (Figure 5c and 5d). Its prediction loss is end-to-end differentiable with respect to the attention generator which leads to efficient training (Section 4.2).

In our theoretical analysis, we show that the attention generator's outputs are invariant to word-substitution perturbations (Section 4.3).

### 4.1  ATTENTION GENERATOR

The goal of the attention generator is to assess word importance from the distributional signatures of each input example. There are many choices of distributional signatures. Among them, we focus on functions of unigram statistics, which are provably robust to word-substitution perturbations (Section 4.3). We utilize the large source pool to inform the model of *general* word importance and leverage the small support set to estimate *class-specific* word importance. The generated attention will be used later to construct the input representation for downstream classification.

---

[2]Note this generated attention can be applied to *any* downstream predictor (Lee et al., 2019; Snell et al., 2017). Details from these experiments may be found in the Appendix A.9.

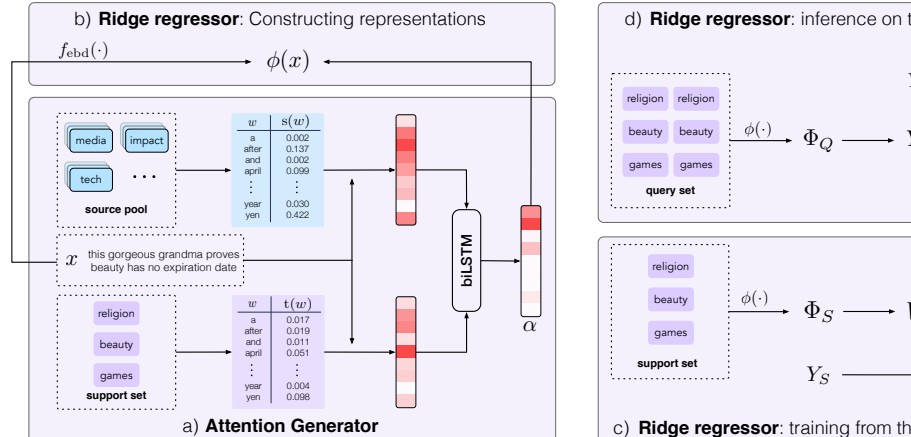

Figure 5: Illustration of our model for an episode with $N = 3$, $K = 1$, $L = 2$. The attention generator translates the distributional signatures from the source pool and the support set into an attention $\alpha$ for each input example $x$ (5a). The ridge regressor utilizes the generated attention to weight the lexical representations (5b). It then learns from the support set (5c) and makes predictions over the query set (5d).

It is well-documented in literature that words which appear frequently are unlikely to be informative (Sparck Jones, 1972). Thus, we would like to downweigh frequent words and upweight rare words. To measure *general word importance*, we select an established approach by Arora et al. (2016):

$$ \mathrm{s}(x_i) := \frac{\varepsilon}{\varepsilon + \mathrm{P}(x_i)} \tag{1} $$

where $\varepsilon = 10^{-3}$, $x_i$ is the $i^{\text{th}}$ word of input example $x$, and $\mathrm{P}(x_i)$ is the unigram likelihood of $x_i$ over the source pool.

On the other hand, words that are discriminative in the support set are likely to be discriminative in the query set. Thus, we define the following statistic to reflect *class-specific word importance*:

$$ \mathrm{t}(x_i) := \mathcal{H}(\mathrm{P}(y \mid x_i))^{-1} \tag{2} $$

where the conditional likelihood $\mathrm{P}(y \mid x_i)$ is estimated over the support set using a regularized linear classifier[3] and $\mathcal{H}(\cdot)$ is the entropy operator. We note that $\mathrm{t}(\cdot)$ measures the uncertainty of the class label $y$, given the word. Thus, words that exhibit a skewed distribution will be highly weighted.

Directly applying these statistics may not result in good performance for two reasons: 1) the two statistics contain complementary information, and it is unclear how to combine them; and 2) these statistics are noisy approximations to word importance for classification. To bridge this gap, we concatenate these signatures and employ a bi-directional LSTM (Hochreiter & Schmidhuber, 1997) to fuse the information across the input: $h = \mathrm{biLSTM}([\mathrm{s}(x); \mathrm{t}(x)])$. Finally, we use dot-product attention to predict the attention score $\alpha_i$ of word $x_i$:

$$ \alpha_i := \frac{\exp\left(v^T h_i\right)}{\sum_j \exp\left(v^T h_j\right)} \tag{3} $$

where $h_i$ is the output of the $\mathrm{biLSTM}$ at position $i$ and $v$ is a learnable vector.

## 4.2 RIDGE REGRESSOR

Informed by the attention generator, the ridge regressor quickly learns to make predictions after seeing a few examples. First, for each example in a given episode, we construct a lexical representation that focuses on important words, as indicated by attention scores. Next, given these lexical representations, we train the ridge regressor on the support set from scratch. Finally, we make predictions over the query set and use the loss to teach the attention generator to produce better attention.

---

[3]See Appendix A.1 for details.

**Constructing representations** Given that different words exhibit varying levels of importance towards classification, we construct lexical representations that favor pertinent words. Specifically, we define the representation of example $x$ as

$$\phi(x) := \sum_i \alpha_i \cdot f_{\text{ebd}}(x_i) \tag{4}$$

where $f_{\text{ebd}}(\cdot)$ is a pre-trained embedding function that maps a word into $\mathbb{R}^E$.

**Training from the support set** Given an $N$-way $K$-shot classification task, let $\Phi_S \in \mathbb{R}^{NK \times E}$ be the representation of the support set, obtained from $\phi(\cdot)$, and $Y_S \in \mathbb{R}^{NK \times N}$ be the one-hot labels. We adopt ridge regression (Bertinetto et al., 2019) to fit the labeled support set for the following reasons: 1) ridge regression admits a closed-form solution that enables end-to-end differentiation through the model, and 2) with proper regularization, ridge regression reduces over-fitting on the small support set. Specifically, we minimize regularized squared loss

$$\mathcal{L}^{RR}(W) := \|\Phi_S W - Y_S\|_F^2 + \lambda \|W\|_F^2 \tag{5}$$

over the weight matrix $W \in \mathbb{R}^{E \times N}$. Here $\|\cdot\|_F$ denotes the Frobenius norm, and $\lambda > 0$ controls the conditioning of the learned transformation $W$. The closed-form solution can be obtained as

$$W = \Phi_S^T (\Phi_S \Phi_S^T + \lambda I)^{-1} Y_S \tag{6}$$

where $I$ is the identity matrix.

**Inference on the query set** Let $\Phi_Q$ denote the representation of the query set. Although we optimized for a regression objective in Eq equation 5, the learned transformation has been shown to work well in few-shot classification after a calibration step (Bertinetto et al., 2019), as

$$\hat{Y}_Q = a\Phi_Q W + b \tag{7}$$

where $a \in \mathbb{R}^+$ and $b \in \mathbb{R}$ are meta-parameters learned through meta-training. Finally, we apply a softmax over $\hat{Y}_Q$ to obtain the predicted probabilities $\hat{P}_Q$. Note that this calibration only adjusts the temperature and scale of the softmax; its mode remains unchanged. During meta-training, we compute the cross-entropy loss $\mathcal{L}^{CE}$ between $\hat{P}_Q$ and the labels over the query set. Since both $\Phi_S$ and $\Phi_Q$ depend on $\phi(\cdot)$, $\mathcal{L}^{CE}$ provides supervision for the attention generator.

### 4.3 THEORETICAL ANALYSIS

Working with distributional signatures brings certified robustness against input perturbations. Formally, let $(\mathcal{P}, \mathcal{S}, \mathcal{Q})$ be a single episode, where $\mathcal{P}$ is the source pool, $\mathcal{S}$ is the support set and $\mathcal{Q}$ is the query set. For any input text $x \in \mathcal{S} \cup \mathcal{Q}$, the attention generator produces word-level importance given the support set and the source pool:

$$\alpha = \text{AttGen}(x \mid \mathcal{S}, \mathcal{P}).$$

Now consider a word-substitution perturbation $\sigma$ that replaces a word $w$ by $\sigma(w)$.[4] If we arbitrarily swap words, we may encounter nonsensical outputs, as important words may be substituted by common words, like "a" or "the." Thus, we consider perturbations that preserve the unigram probabilities of words, estimated over the source pool: $P(w) = P(\sigma(w))$ for all $w \in V$. In this way, important words (for one class) are mapped to similarly important words (perhaps for another class). We use $\tilde{\mathcal{S}}$ and $\tilde{\mathcal{Q}}$ to denote the support and query sets after applying $\sigma$ word by word.

**Theorem 1.** *Assume $\sigma : V \to V$ satisfies $P(w) = P(\sigma(w))$ for all $w$. If $\sigma$ is a bijection, then for any input text $x \in \mathcal{S} \cup \mathcal{Q}$ and its perturbation $\tilde{x} \in \tilde{\mathcal{S}} \cup \tilde{\mathcal{Q}}$, the outputs of the attention generator are the same:*

$$\text{AttGen}(x \mid \mathcal{S}, \mathcal{P}) = \text{AttGen}(\tilde{x} \mid \tilde{\mathcal{S}}, \mathcal{P}).$$

The idea behind this proof is to show that general word importance $s$ and class-specific word importance $t$ are invariant to such perturbations. Details may be found in Appendix A.2.

Alternatively, one can view this word-substitution perturbation as a change to the input distribution. Let $P(x \mid y)$ be the conditional probability of the input $x$ given the class label $y$. Assuming that $\sigma$

---

[4]We view this $\sigma$ as a mapping from the vocabulary set $V$ to itself.

is a bijection, the conditional after perturbation can be expressed as $\tilde{P}(x \mid y) = P\left(\sigma^{-1}(x) \mid y\right)$, where $\sigma^{-1}(x)$ is the result of applying $\sigma^{-1}$ to every word in $x$. Intuitively, Theorem 1 tells us that if a word $w$ is a discriminative feature for the original classification task, then after the change of the input distribution, the word $\sigma(w)$ should be a discriminative feature for the new task. This property makes sense since word-importance for classification should depends only on the relative differences between classes. In fact, the theorem holds when the input to the attention generator is any function of unigram counts. We also note that the classification performance on the query set can be different, as the pre-trained embedding function $f_{\text{ebd}}(\cdot)$ is generally not invariant under such perturbations.

## 5   EXPERIMENTAL SETUP

### 5.1   DATASETS

We evaluate our approach on five text classification datasets and one relation classification dataset.[5] (See Appendix A.4 for more details.)

**20 Newsgroups**  is comprised of informal discourse from news discussion forums (Lang, 1995). Documents are organized under 20 topics.

**RCV1**  is a collection of Reuters newswire articles from 1996 to 1997 (Lewis et al., 2004). These articles are written in formal speech and labeled with a set of topic codes. We consider 71 second-level topics as our total class set and discard articles that belong to more than one class.

**Reuters-21578**  is a collection of shorter Reuters articles from 1987 (Lewis, 1997). We use the standard ApteMod version of the dataset. We discard articles with more than one label and consider 31 classes that have at least 20 articles.

**Amazon product data**  contains customer reviews from 24 product categories (He & McAuley, 2016). Our goal is to classify reviews into their respective product categories. Since the original dataset is notoriously large (142.8 million reviews), we select a more tractable subset by sampling 1000 reviews from each category.

**HuffPost headlines**  consists of news headlines published on HuffPost between 2012 and 2018 (Misra, 2018). These headlines split among 41 classes. They are shorter and less grammatical than formal sentences.

**FewRel**  is a relation classification dataset developed for few-shot learning (Han et al., 2018). Each example is a single sentence, annotated with a head entity, a tail entity, and their relation. The goal is to predict the correct relation between the head and tail. The public dataset contains 80 relation types.

### 5.2   BASELINES

We compare our model (denoted as OUR) to different combinations of representations and learning algorithms. Details of the baselines may be found in Appendix A.11.

**Representations**  We evaluate three representations. AVG represents each example as the mean of its embeddings. IDF represents each example as the weighted average of its word embeddings, with weights given by inverse document frequency over all training examples. CNN applies 1D convolution over the input words and obtains the representation by max-over-time pooling (Kim, 2014).

**Learning algorithms**  In addition to the ridge regressor (RR) (Bertinetto et al., 2019), we evaluate two standard supervised learning algorithms and two meta-learning algorithms. NN is a 1-nearest-neighbor classifier under Euclidean distance. FT pre-trains a classifier over all training examples, then finetunes the network using the support set (Chen et al., 2019). MAML meta-learns a prior over model parameters, so that the model can quickly adapt to new classes (Finn et al., 2017). Prototypical network (PROTO) meta-learns a metric space for few-shot classification by minimizing

---

[5]All processed datasets along with their splits are publicly available.

| Method | | 20 News | | Amazon | | HuffPost | | RCV1 | | Reuters | | FewRel | | Average | |
|---|---|---|---|---|---|---|---|---|---|---|---|---|---|---|---|---|
| Rep. | Alg. | 1 shot | 5 shot | 1 shot | 5 shot | 1 shot | 5 shot | 1 shot | 5 shot | 1 shot | 5 shot | 1 shot | 5 shot | 1 shot | 5 shot |
| AVG | NN | 33.9 | 45.8 | 46.7 | 60.3 | 31.4 | 41.5 | 43.7 | 60.8 | 56.5 | 80.5 | 47.5 | 60.6 | 43.3 | 58.2 |
| IDF | NN | 38.8 | 51.9 | 51.4 | 67.1 | 31.5 | 42.3 | 41.9 | 58.2 | 57.8 | 82.9 | 46.8 | 60.6 | 44.7 | 60.5 |
| CNN | FT | 33.0 | 47.1 | 45.7 | 63.9 | 32.4 | 44.1 | 40.3 | 62.3 | 70.9 | 91.0 | 54.0 | 71.1 | 46.0 | 63.2 |
| AVG | PROTO | 36.2 | 45.4 | 37.2 | 51.9 | 35.6 | 41.6 | 28.4 | 31.2 | 59.5 | 68.1 | 44.0 | 46.5 | 40.1 | 47.4 |
| IDF | PROTO | 37.8 | 46.5 | 41.9 | 59.2 | 34.8 | 50.2 | 32.1 | 35.6 | 61.0 | 72.1 | 43.0 | 61.9 | 41.8 | 54.2 |
| CNN | PROTO | 29.6 | 35.0 | 34.0 | 44.4 | 33.4 | 44.2 | 28.4 | 29.3 | 65.2 | 74.3 | 49.7 | 65.1 | 40.1 | 48.7 |
| AVG | MAML | 33.7 | 43.9 | 39.3 | 47.2 | 36.1 | 49.6 | 39.9 | 50.6 | 54.6 | 62.5 | 43.8 | 57.8 | 41.2 | 51.9 |
| IDF | MAML | 37.2 | 48.6 | 43.6 | 62.4 | 38.9 | 53.7 | 42.5 | 54.1 | 61.5 | 72.0 | 48.2 | 65.8 | 45.3 | 59.4 |
| CNN | MAML | 28.9 | 36.7 | 35.3 | 43.7 | 34.1 | 45.8 | 39.0 | 51.1 | 66.6 | 85.0 | 51.7 | 66.9 | 42.6 | 54.9 |
| AVG | RR | 37.6 | 57.2 | 50.2 | 72.7 | 36.3 | 54.8 | 48.1 | 72.6 | 63.4 | 90.0 | 53.2 | 72.2 | 48.1 | 69.9 |
| IDF | RR | 44.8 | 64.3 | 60.2 | 79.7 | 37.6 | 59.5 | 48.6 | 72.8 | 69.1 | 93.0 | 55.6 | 75.3 | 52.6 | 74.1 |
| CNN | RR | 32.2 | 44.3 | 37.3 | 53.8 | 37.3 | 49.9 | 41.8 | 59.4 | 71.4 | 87.9 | 56.8 | 71.8 | 46.1 | 61.2 |
| OUR | | **52.1** | **68.3** | **62.6** | **81.1** | **43.0** | **63.5** | **54.1** | **75.3** | **81.8** | **96.0** | **67.1** | **83.5** | **60.1** | **78.0** |
| OUR w/o t(·) | | 50.1 | 67.5 | 61.7 | 80.5 | 42.0 | 60.8 | 51.5 | 75.1 | 76.7 | 93.7 | 66.9 | 83.2 | 58.1 | 76.8 |
| OUR w/o s(·) | | 41.9 | 60.7 | 51.1 | 75.3 | 40.1 | 60.2 | 48.5 | 72.8 | 78.1 | 94.8 | 65.8 | 82.6 | 54.2 | 74.4 |
| OUR w/o biLSTM | | 50.3 | 66.9 | 61.9 | 80.9 | 42.2 | 63.0 | 51.8 | 74.1 | 77.2 | 95.4 | 66.4 | 82.9 | 58.3 | 77.2 |
| OUR w EBD | | 39.7 | 57.5 | 56.5 | 76.3 | 40.6 | 58.6 | 48.6 | 71.5 | 81.7 | 95.8 | 61.5 | 80.9 | 54.8 | 73.4 |

Table 1: Results of 5-way 1-shot and 5-way 5-shot classification on six datasets. The bottom four rows present our ablation study. For complete results with standard deviations see Table 8 and 9 in Appendix A.12.

the Euclidean distance between the centroid of each class and its constituent examples (Snell et al., 2017).

## 5.3 IMPLEMENTATION DETAILS

We use pre-trained fastText embeddings (Joulin et al., 2016) for our model and all baselines. For sentence-level datasets (HuffPost, FewRel), we also experiment with pre-trained BERT embeddings (Devlin et al., 2018) using HuggingFace's codebase (Wolf et al., 2019). For relation classification (FewRel), we augment the input of our attention generator with positional embeddings (Zhang et al., 2017).[6]

In the attention generator, we use a $\mathrm{biLSTM}$ with 50 hidden units and apply dropout of 0.1 on the output (Srivastava et al., 2014). In the ridge regressor, we optimize meta-parameters $\lambda$ and $a$ in the $\log$ space to maintain the positivity constraint. All parameters are optimized using Adam with a learning rate of 0.001 (Kingma & Ba, 2014).

During meta-training, we sample 100 training episodes per epoch. We apply early stopping when the validation loss fails to improve for 20 epochs. We evaluate test performance based on 1000 testing episodes and report the average accuracy over 5 different random seeds.

## 6 RESULTS

We evaluated our model in both 5-way 1-shot and 5-way 5-shot settings. These results are reported in Table 1. Our model consistently achieves the best performance across all datasets. On average, our model improves 5-way 1-shot accuracy by 7.5% and 5-way 5-shot accuracy by 3.9%, against the best baseline for each dataset. When comparing against CNN+PROTO, our model improves by 20.0% on average in 1-shot classification. The empirical results clearly demonstrate that meta-learners privy to lexical information consistently fail, while our model is able to generalize past class-specific vocabulary. Furthermore, Figure 6 illustrates that a lexicon-aware meta-learner (CNN+PROTO) is able to overfit the training data faster than our model, but our model more readily generalizes to unseen classes.

---

[6]We also provide the same positional embeddings to the baseline CNN.

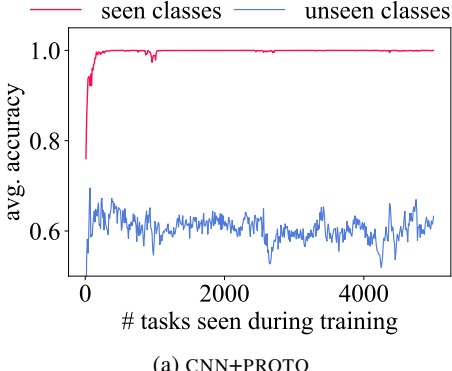

(a) CNN+PROTO

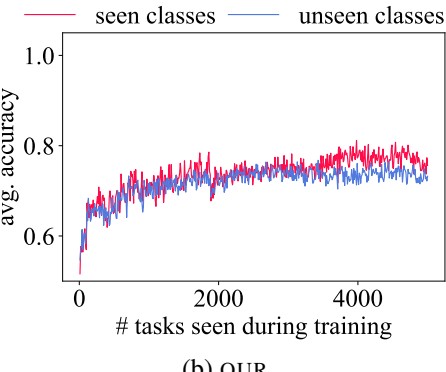

(b) OUR

Figure 6: Learning curve of CNN+PROTO (left) v.s. OUR (right) on the Reuters dataset. We plot average 5-way 1-shot accuracy over 50 episodes sampled from seen classes (blue) and unseen classes (red). While OUR has weaker representational power, it generalizes better to unseen classes.

| Method | | HuffPost | | FewRel | |
|---|---|---|---|---|---|
| Rep. | Alg. | 1 shot | 5 shot | 1 shot | 5 shot |
| AVG | NN | $23.23_{\pm 0.20}$ | $34.22_{\pm 0.28}$ | $52.37_{\pm 0.27}$ | $64.85_{\pm 0.25}$ |
| IDF | NN | $33.49_{\pm 0.18}$ | $45.71_{\pm 0.17}$ | $48.65_{\pm 0.18}$ | $62.35_{\pm 0.18}$ |
| CNN | FT | $37.30_{\pm 1.08}$ | $51.56_{\pm 1.28}$ | $61.10_{\pm 2.54}$ | $80.04_{\pm 0.69}$ |
| AVG | PROTO | $34.21_{\pm 0.56}$ | $49.77_{\pm 1.90}$ | $50.27_{\pm 0.98}$ | $66.24_{\pm 2.01}$ |
| IDF | PROTO | $36.06_{\pm 0.84}$ | $54.58_{\pm 0.99}$ | $48.23_{\pm 0.58}$ | $67.82_{\pm 0.72}$ |
| CNN | PROTO | $36.17_{\pm 1.00}$ | $50.55_{\pm 0.96}$ | $57.08_{\pm 5.52}$ | $75.01_{\pm 2.21}$ |
| AVG | MAML | $38.58_{\pm 1.56}$ | $55.32_{\pm 1.42}$ | $47.18_{\pm 3.49}$ | $64.50_{\pm 2.72}$ |
| IDF | MAML | $34.22_{\pm 0.74}$ | $56.50_{\pm 1.50}$ | $50.06_{\pm 2.88}$ | $68.43_{\pm 2.50}$ |
| CNN | MAML | $38.39_{\pm 1.68}$ | $53.86_{\pm 0.76}$ | $47.68_{\pm 1.66}$ | $71.56_{\pm 4.75}$ |
| AVG | RR | $25.34_{\pm 0.14}$ | $51.52_{\pm 0.14}$ | $55.65_{\pm 0.27}$ | $73.91_{\pm 0.77}$ |
| IDF | RR | $40.38_{\pm 0.11}$ | $61.72_{\pm 1.03}$ | $54.48_{\pm 0.26}$ | $73.48_{\pm 0.72}$ |
| CNN | RR | $41.37_{\pm 0.54}$ | $53.10_{\pm 0.76}$ | $65.65_{\pm 5.70}$ | $78.65_{\pm 4.24}$ |
| OUR | | $\mathbf{42.12}_{\pm 0.15}$ | $\mathbf{62.97}_{\pm 0.67}$ | $\mathbf{70.08}_{\pm 0.56}$ | $\mathbf{88.07}_{\pm 0.27}$ |

Table 2: 5-way 1-shot and 5-way 5-shot classification on HuffPost and FewRel using BERT.

**Ablation study** We perform ablation studies on the attention generator. These results are shown at the bottom of Table 1. We observe that both statistics $s(\cdot)$ and $t(\cdot)$ contribute to the performance, though the former has a larger impact. We also note that instead of computing word importance independently for each word using a multilayer perceptron (OUR w/o biLSTM), fusing information across the input with an biLSTM improves performance slightly.

Finally, we observe that restricting meta-knowledge to distributional signatures is essential to performance: when both lexical word embeddings and distributional signatures are fed to the attention generator (OUR w EBD), the performance drops consistently.

**Contextualized representations** For sentence-level datasets (FewRel, HuffPost), we also experiment with contextualized representations, given by BERT (Devlin et al., 2018). These results are shown in Table 2. While BERT significantly improves classification performance on FewRel, we observe no performance boost on HuffPost. We postulate that this discrepancy arises because relation classification is highly contextual, while news classification is mostly keyword-based.

**Analysis** We visualize our attention-weighted representation $\phi(x)$ in Figure 7. Compared to directly using general word importance $s(x)$ or class-specific word importance $t(x)$, our method produces better separation, which enables effective learning from a few examples. This highlights that the power of our approach lies not in the distributional signatures themselves, but rather in the representations learned on top of them.

Figure 8 visualizes the model's input and output on the same query example in two testing episodes. The example belongs to the class *jobs* in the Reuters dataset. First, we observe that our model generates meaningful attention from noisy distributional signatures. Furthermore, the generated

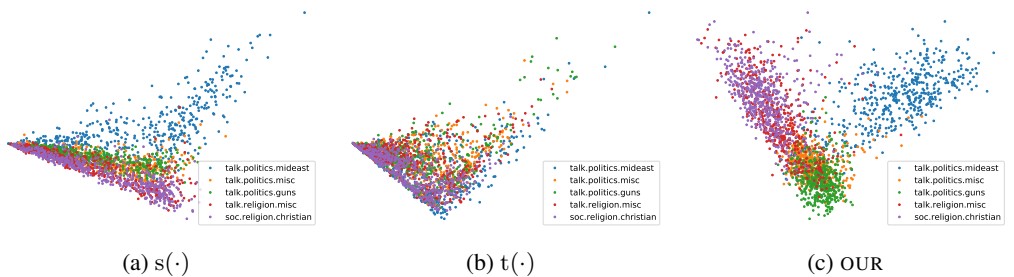

(a) s(·)  (b) t(·)  (c) OUR

Figure 7: PCA visualization of the input representation for the query set of a testing episode ($N = 5$, $K = 5$, $L = 500$) sampled from 20 Newsgroups. Weighted averages of word embeddings given by (a) s(·), (b) t(·), and (c) the attention generator meta-trained on a disjoint set of training classes.

| | Fine-grained classification | Coarse-grained classification |
|---|---|---|
| $s(x)$ | finnish unemployment was 6.7 pct in december last year compared with 6.8 pct in november and 6.1 pct in december 1985 , the central statistical office said | finnish unemployment was 6.7 pct in december last year compared with 6.8 pct in november and 6.1 pct in december 1985 , the central statistical office said |
| $t(x)$ | finnish unemployment was 6.7 pct in december last year compared with 6.8 pct in november and 6.1 pct in december 1985 , the central statistical office said | finnish unemployment was 6.7 pct in december last year compared with 6.8 pct in november and 6.1 pct in december 1985 , the central statistical office said |
| OURS | finnish unemployment was 6.7 pct in december last year compared with 6.8 pct in november and 6.1 pct in december 1985 , the central statistical office said | finnish unemployment was 6.7 pct in december last year compared with 6.8 pct in november and 6.1 pct in december 1985 , the central statistical office said |

Figure 8: Attention weights generated by our model are specific to task. We visualize our model's inputs s($x$) (top), t($x$) (middle), and output (bottom) for one query example from class *jobs* in Reuters dataset. Word "statistical" is downweighed for *jobs* when compared to other economics classes (left), but it becomes important when considering dissimilar classes (right). Fine-grained classes: *jobs*, *retail*, *industrial production index*, *wholesale production index*, *consumer production index*. Coarse-grained classes: *jobs*, *cocoa*, *aluminum*, *copper*, *reserves*.

attention is *task-specific*: in the depicted example, if the episode contains other economics-related classes, the word "statistical" is downweighed by our model. Conversely, "statistical" is upweighted when we compare *jobs* to other distant classes.

## 7 CONCLUSION

In this paper, we propose a novel meta-learning approach that capitalizes on the connection between word importance and distributional signatures to improve few-shot classification. Specifically, we learn an attention generator that translates distributional statistics into high-quality attention. This generated attention then provides guidance for fast adaptation to new classification tasks. Experimental results on both text and relation classification validate that our model identifies important words for new classes. The effectiveness of our approach demonstrates the promise of meta learning with distributional signatures.

## ACKNOWLEDGMENTS

We thank the MIT NLP group and the reviewers for their helpful discussion and comments. This work is supported by MIT-IBM Watson AI Lab and Facebook Research. Any opinions, findings, conclusions, or recommendations expressed in this paper are those of the authors, and do not necessarily reflect the views of the funding organizations.

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

# A    Supplemental Material

## A.1    Regularized linear classifier

Given an $N$-way $K$-shot classification task, the goal of the regularized linear classifier is to approximate task-specific word importance using the support set.

Let $x = \{x_1, \ldots x_T\}$ be an input example, and let $f_{\mathrm{ebd}}(\cdot)$ be an embedding function that maps each word $x_i$ into $\mathbb{R}^E$. We compute the representation of $x$ by the average of its embeddings:

$$\psi(x) := \frac{1}{T} \sum_i f_{\mathrm{ebd}}(x_i).$$

Since the support set only contains a few examples, we adopt a simple linear classifier to reduce overfitting:

$$\hat{p} := \mathrm{softmax}(W\psi(x))$$

where $W \in \mathbb{R}^{N \times E}$ is the weight matrix to learn. We minimize the cross entropy loss between the prediction $\hat{p}$ and the ground truth label while penalizing the Frobenius norm of $W$. We stop training once the gradient norm is less than $0.1$. Finally, given a word $x_i$, we estimate its conditional probability via $\mathrm{softmax}(W\psi(x_i))$.

**Time efficiency**  Since the support set is very small (less than 25 examples) and the loss function is strongly convex, this linear classifier converges very fast in practice.[7] Note that for larger problems, we can speed up computation by formulating this procedure as a regression problem and solving for its closed-form solution (as in Section 4.2).

## A.2    Proof of Theorem 1

**Theorem 1.** *Assume $\sigma : V \to V$ satisfies $\mathrm{P}(w) = \mathrm{P}(\sigma(w))$ for all $w$. If $\sigma$ is a bijection, then for any input text $x \in \mathcal{S} \cup \mathcal{Q}$ and its perturbation $\tilde{x} \in \tilde{\mathcal{S}} \cup \tilde{\mathcal{Q}}$, the outputs of the attention generator are the same:*

$$AttGen(x \mid \mathcal{S}, \mathcal{P}) = AttGen(\tilde{x} \mid \tilde{\mathcal{S}}, \mathcal{P}).$$

*Proof.* It suffices to show that the general word importance $\mathrm{s}(\cdot)$ and the class-specific word importance $\mathrm{t}(\cdot)$ are the same under the perturbation $\phi$. For the former, since $\phi$ preserves unigram probability estimated over the source pool, we have

$$\mathrm{s}(x_i) = \frac{\varepsilon}{\varepsilon + \mathrm{P}(x_i)} = \frac{\varepsilon}{\varepsilon + \mathrm{P}(\sigma(x_i))} = \mathrm{s}(\sigma(x_i)).$$

To prove the latter, we need to show that the conditional probability $\mathrm{P}(y \mid x_i)$ estimated over the support set is also invariant under $\phi$. Let $\#(x_i \wedge y \mid \mathcal{S})$ denote the occurrences of the word $x_i$ in examples with class label $y$ over the support set $\mathcal{S}$. Using maximum likelihood estimation, we have

$$\hat{\mathrm{P}}(y \mid x_i, \mathcal{S}) = \frac{\#(x_i \wedge y \mid \mathcal{S})}{\sum_{y'} \#(x_i \wedge y' \mid \mathcal{S})} = \frac{\#(\sigma(x_i) \wedge y \mid \tilde{\mathcal{S}})}{\sum_{y'} \#(\sigma(x_i) \wedge y' \mid \tilde{\mathcal{S}})} = \hat{\mathrm{P}}(y \mid \sigma(x_i), \tilde{\mathcal{S}}),$$

where the second equality is derived from the fact that $\sigma$ is a bijection.  $\square$

## A.3    Learning procedure

Algorithm 1 contains the pseudo code for our learning procedure. We apply early stopping when validation loss does not improve for 20 epochs.

## A.4    Datasets

To reliably test our model's ability to generalize across classes, we consider two data splitting mechanisms in our experiments: 1) *easy split*: we randomly permute all classes and split them into train/val/test; 2) *hard split*: we select train/val/test based on the class hierarchy such that train

---

[7]less than 1 second on a single GeForce GTX TITAN X

**Algorithm 1** Meta-training procedure. $N_{\text{train}} = \left|\mathcal{Y}^{\text{train}}\right|$ is the number of training classes. $N < N_{\text{train}}$ is the number of classes of each few-shot task. $K, L$ are the number of support and query examples per target class, respectively. To ease notation, we use $\text{SAMPLE}(S, N)$ to denote a subset of $S$ with $N$ elements, chosen uniformly at random. We use $\mathcal{D}(\mathcal{Y}) \subseteq \mathcal{D}$ to denote the set of all elements $(x_i, y_i)$ for which $y_i \in \mathcal{Y}$.

---

**Input:** Training set $\mathcal{D} = \{(x_1, y_1), (x_2, y_2), \ldots\}$ where each $y_i \in \mathcal{Y}^{\text{train}} = \{1, \ldots, N_{\text{train}}\}$
**Hyperparameters:** parameters of $\phi$; coefficients for losses $\lambda, a, b$
  **repeat**
      $\mathcal{Y} \leftarrow \text{SAMPLE}(\mathcal{Y}^{\text{train}}, N)$                                        ▷ sample target classes
      $\mathcal{D}^{\text{src}} \leftarrow \mathcal{D}(\mathcal{Y}^{\text{train}} \setminus \mathcal{Y})$                              ▷ create support pool
      $\mathcal{T}^{\text{S}}, \mathcal{T}^{\text{Q}} \leftarrow \emptyset, \emptyset$                           ▷ sample support set and query set
      **for** $y \in \mathcal{Y}$ **do**
         $\mathcal{T}^{\text{S}} \leftarrow \mathcal{T}^{\text{S}} \cup \text{SAMPLE}(\mathcal{D}(\{y\}), K)$
         $\mathcal{T}^{\text{Q}} \leftarrow \mathcal{T}^{\text{Q}} \cup \text{SAMPLE}(\mathcal{D}(\{y\}) \setminus \mathcal{T}^{\text{S}}, L)$
      Generate attention scores using $\mathcal{T}^{\text{S}}, \mathcal{T}^{\text{Q}}, \mathcal{D}^{\text{src}}$              ▷ Eq. 3
      Construct attention-weighted representations $\Phi_{\text{S}}, \Phi_{\text{Q}}$ using $\phi(\cdot)$     ▷ Eq. 4
      Compute closed-form solution $W$ of ridge regression from $\lambda, \Phi_S$, and support labels $Y_{\text{S}}$   ▷ Eq. 6
      Given $W, a, b$, compute cross entropy loss $\mathcal{L}^{\text{CE}}$ on the query set $\Phi_{\text{Q}}, Y_{\text{Q}}$
      Update meta parameters (attention generator and $a, B$) using $\mathcal{L}^{\text{CE}}$
  **until** stopping criterion is met

---

| Dataset | # tokens / example | vocab size | # examples / cls | # train cls | # val cls | # test cls |
|---|---|---|---|---|---|---|
| 20 Newsgroups | 340 | 32137 | 941 | 8 | 5 | 7 |
| RCV1 | 372 | 7304 | 20 | 37 | 10 | 24 |
| Reuters | 168 | 2234 | 20 | 15 | 5 | 11 |
| Amazon | 140 | 17062 | 1000 | 10 | 5 | 9 |
| HuffPost | 11 | 8218 | 900 | 20 | 5 | 16 |
| FewRel | 24 | 16045 | 700 | 65 | 5 | 10 |

Table 3: Dataset statistics. See Appendix A.4 for details.

classes are distant to val and test. We applied the easy split to one sentence-level dataset (Huff-Post) and one document-level dataset (Reuters-21578). Hard split is used for the other four datasets (details below). This setting tests the generalization capacity of the algorithm, following Xian et al. (2017).

**20 Newsgroups** Each class in 20 Newsgroups belongs to one of six top-level categories, which roughly correspond to computers, recreation, science, politics, religion, and for-sale. We partition the set of labels so that no top-level category spans two splits. Train contains "sci" and "rec," val contains "comp," and test contains all other labels.

**Amazon** The Amazon dataset does not come with predefined top-level categories. To generate a hard split, we first apply spectral clustering to classes based on their word distributions. Then we select train/val/test from different clusters.

**RCV1** We apply the same approach as above.

**FewRel** While FewRel does not provide higher-level categories, we observe that most relations occur between named entities of similar types. Thus, we extract the named entity type of the head and tail for each example using a pretrained spaCy parser.[8] For each class, we determine the most common head and tail entity types. Test contains all classes for which the most common head entity type is WORK_OF_ART. Train and validation were arbitrarily split to contain the remaining relations.

## A.5 OTHER BASELINES

We also consider two other baselines: induction network (Geng et al., 2019) and P-MAML (Zhang et al., 2019). Implementation details may be found in Appendix A.11.

---

[8]https://spacy.io/

| Method | 20 News | | Amazon | | HuffPost | | RCV1 | | Reuters | | FewRel | |
|---|---|---|---|---|---|---|---|---|---|---|---|---|
| | 1 shot | 5 shot | 1 shot | 5 shot | 1 shot | 5 shot | 1 shot | 5 shot | 1 shot | 5 shot | 1 shot | 5 shot |
| INDUCTION NET | 27.6 | 32.1 | 30.6 | 37.1 | 34.9 | 44.0 | 32.3 | 37.3 | 58.3 | 66.9 | 50.4 | 56.1 |
| P-MAML | — | — | 47.1 | 58.4 | 31.0 | 51.3 | — | — | 53.0 | 72.9 | — | — |
| OUR | **52.1** | **68.3** | **62.6** | **81.1** | **43.0** | **63.5** | **54.1** | **75.3** | **81.8** | **96.0** | **67.1** | **83.5** |

Table 4: Comparison against Induction Net (Geng et al., 2019) and P-MAML (Zhang et al., 2019). We run P-MAML on text classification datasets with shorter documents, for which it is feasible to finetune BERT (after WordPiece tokenization, documents from "longer" datasets exceed BERT's max length of 512 tokens).

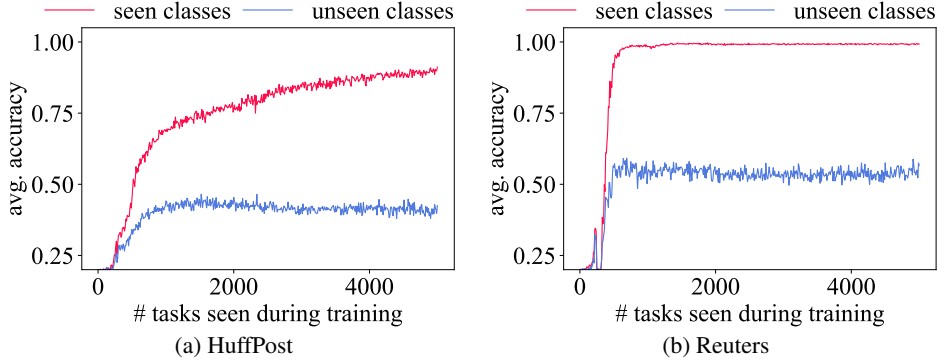

(a) HuffPost
(b) Reuters

Figure 9: Learning curves of INDUCTION NET on Huffpost and Reuters. We plot average 5-way 5-shot accuracy over 50 episodes sampled from seen classes (blue) and unseen classes (red).

- **Induction network** encodes input examples using a biLSTM with self-attentive pooling (Lin et al., 2017). Based on the encoded representation, it then computes a prototype for each class through dynamic routing over the support set (Sabour et al., 2017). Finally, it uses a neural tensor layer (Socher et al., 2013) to predict the relation between each query example and the class prototypes.

- **P-MAML** combines pre-training with MAML. It first finetunes pre-trained BERT representation on the meta-training data using masked language modeling (Devlin et al., 2018). Based on this finetuned representation, it trains first-order MAML Finn et al. (2017) to enable fast adaptation.

Table 4 shows that learning with distributional signatures (OUR) significantly outperforms the two baselines across all datasets. This is not surprising: both baselines build meta-knowledge on top of lexical representations, which may not generalize when the lexical distributions are vastly different between seen classes and unseen classes. Figure 9 shows the learning curve of INDUCTION NET. Similar to Figure 6, we see that INDUCTION NET overfits to meta-train classes with poor generalization to meta-test classes. Figure 10 depicts the perplexity of BERT's masked language model objective, during finetuning for P-MAML. Again, we see that the meta-train and meta-test classes exhibit vast lexical mismatch, as the perplexity improves only slightly on meta-test classes.

### A.6 ANALYSIS OF LEARNED REPRESENTATION

To further understand the rationale behind our performance boost, we provide a more detailed analysis on our model's empirical behavior.

**Visualizing the embedding space** We visualize our attention-weghted representation $\phi(x)$ in 20 Newsgroups (Figure 11) and HuffPost (Figure 12). We observe that our model produces better separation than the unweighted average AVG and directly using the distributional statistics, $s(x)$ or $t(x)$. For instance, in 20 Newsgroups, our model recognizes three clusters: {*talk.religion.misc, soc.religion.christian*}, {*talk.politics.mideast*} and {*talk.politics.misc, talk.politics.guns*},

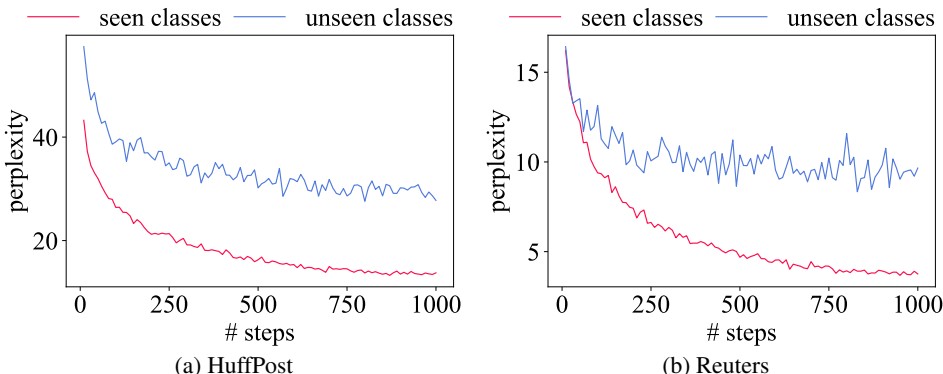

Figure 10: Perplexity during BERT language model finetuning in P-MAML. The lexical distributions mismatch significantly between meta-train and meta-test classes.

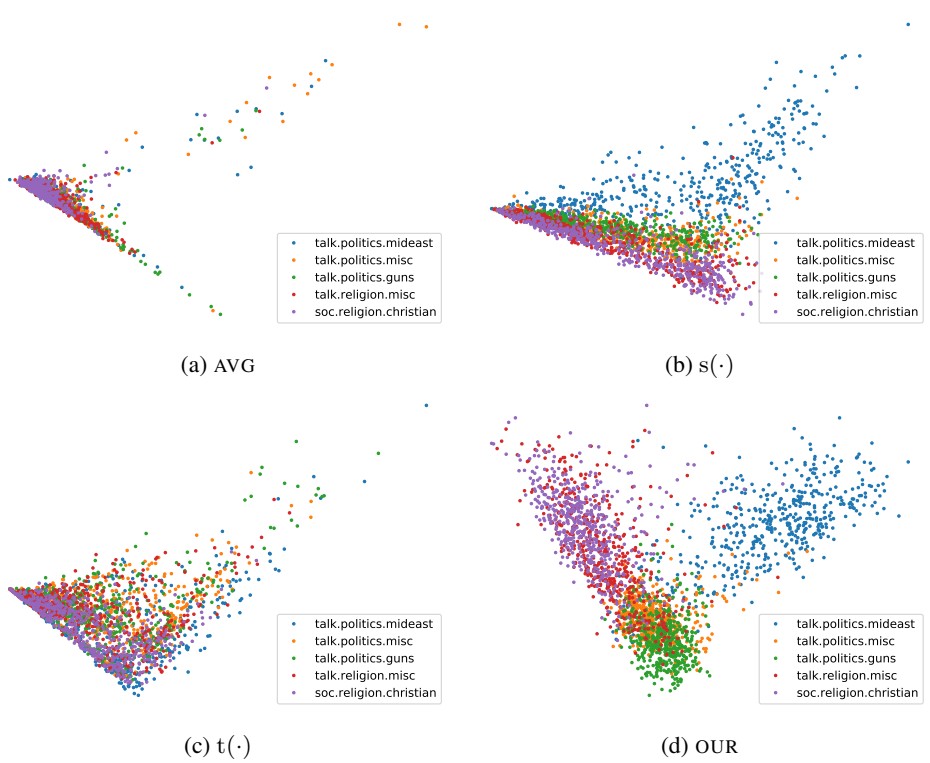

Figure 11: PCA visualization of the input representation for a testing episode in 20 Newsgroups with $N = 5$, $K = 5$, $L = 500$ (the query set has 500 examples per class). AVG: average word embeddings. $s(\cdot)$: weighted average of word embeddings with weights given by $s(\cdot)$. $t(\cdot)$: weighted average of word embeddings with weights given by $t(\cdot)$. OUR: weighted average of word embeddings with weights given by the attention generator meta-trained on a disjoint set of training classes.

**Cosine similarity to *oracle* word importance** We also quantitatively analyze the generated attention in in 20 Newsgroups and HuffPost (Figure 13).

To obtain a more reliable estimate of word importance, we train an *oracle* model over all examples from the $N$ target classes. This oracle model uses a biLSTM to encode each example from its word embeddings. It then generates an attention score based on this encoding. In order to estimate the importance of individual unigram, we use this attention score to weight the "original" word

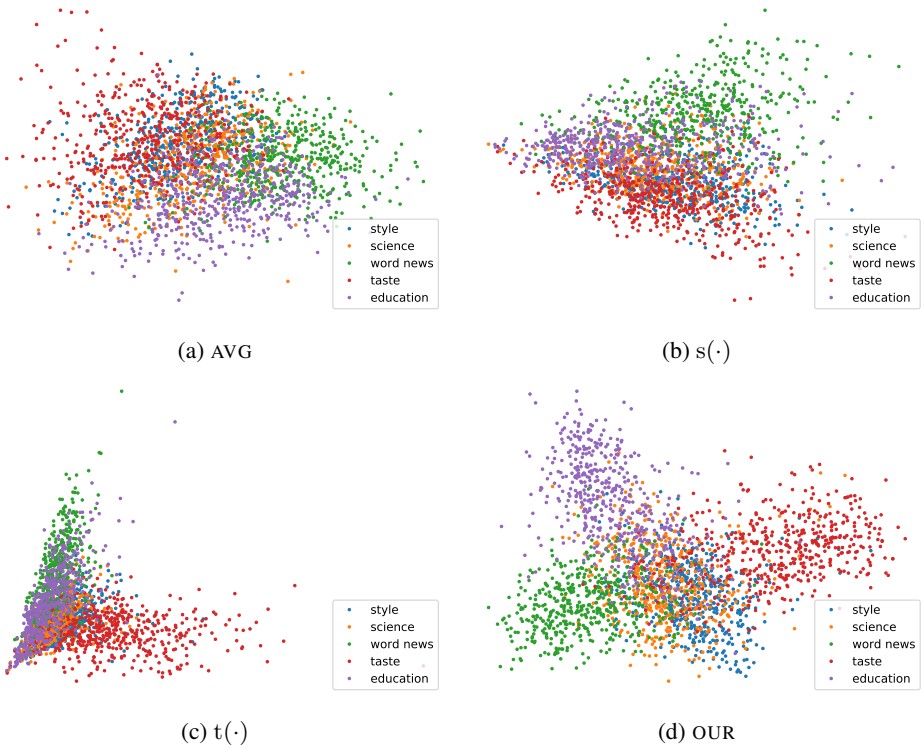

(a) AVG
(b) s(·)

(c) t(·)
(d) OUR

Figure 12: PCA visualization of the input representation for a testing episode in HuffPost Headlines with $N = 5$, $K = 5$, $L = 500$. AVG: average word embeddings. s(·): weighted average of word embeddings with weights given by s(·). t(·): weighted average of word embeddings with weights given by t(·). OUR: weighted average of word embeddings with weights given by the attention generator meta-trained on a disjoint set of training classes.

embeddings (not the output of the biLSTM). A MLP with one hidden layer is used to perform $N$-way classification from the attention-weighted representation.

Compared to both s(·) and t(·), the attention generated by our model is much closer to the oracle's, which explains our model's large performance gains. Note that the attention generator does not see any examples from the target classes during meta-training.

**Visualizing the generated attention** Figure 14 visualizes the generated attention for a testing episode in HuffPost. We observe that our model identifies meaningful keywords from the sentence.

## A.7 EFFECT OF SOFTMAX CALIBRATION

In Section 4.2, we use ridge regression with softmax calibration as our downstream predictor due to its efficiency and effectiveness. To study the effect of the softmax calibration, we follow Bertinetto et al. (2019) and compare ridge regression against direct optimization of binary logistic classifiers (LR) with Newton's method (five iterations). Since Newton's method admits a closed-form solution at each iteration, the final solution is also end-to-end differentiable. For each $N$-way $K$-shot episode, to adapt the binary logistic classifier for multi-class prediction, we train $N$ binary (one-vs-all) classifiers using the support set. The output of the $N$ binary classifiers are combined to make the final predictions on the query set (Bishop, 2006, Chapter 4.1.2).

The results are shown in Table 5. Overall, RR and LR produce similar results (78.0 vs. 77.1 on 5-way 5-shot, 60.1 vs. 58.5 on 5-way 1-shot, respectively), though RR performs slightly better. When using LR as the downstream predictor, learning with distributional signatures improves 5-way 1-shot accuracy by 6.1% and 5-way 5-shot accuracy by 3.2% over the best baseline. Note that LR requires solving $N$ binary classifications for each episode, which is less efficient than RR.

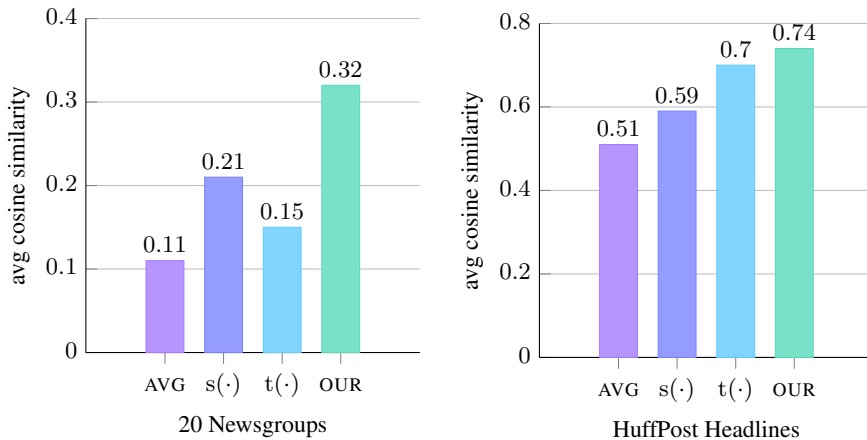

Figure 13: Average cosine similarity to the *oracle* word importance over the query set of a testing episode with $N = 5$, $K = 5$, $L = 500$. This oracle is estimated using all labeled examples from the $N$ classes. Since examples in HuffPost Headlines are 30 times shorter, the cosine similarities are higher in this corpora. AVG: uniform distribution over the words. s($\cdot$): word importance estimated directly by s($\cdot$). t($\cdot$): word importance estimated directly by t($\cdot$). OUR: word importance estimated by the meta-learned attention generator.

| class | input example |
|---|---|
| taste | you wo n't even miss the meat with these delicious vegetarian sandwiches |
| taste | these cookies are spot - on copies of the oscars dresses |
| word news | prime minister saad hariri 's return to lebanon : a moment of truth |
| word news | new zealand just became the 11th country to send a rocket into orbit |
| style | beyoncé dressed like the queen she is at the grammys |
| style | tilda swinton , is that a jacket or a dress ? |
| science | the world of science has a lot to look forward to in 2016 |
| science | dione crosses saturn 's disk in spectacular new image |
| education | the global search for education : just imagine secretary hargreaves |
| education | thinking at harvard : what is the future of learning ? |

Figure 14: Visualization of the attention generated by our model on 10 query examples from a 5-way 5-shot testing episode in Huffpost Headlines.

## A.8 FINETUNING WORD EMBEDDINGS DURING META-TRAINING

In Table 1, we fix word embeddings during meta-training for all experiments. Table 6 studies the effect of finetuning word embeddings during meta-training. We observe that when the word embeddings are finetuned, performance drops for nearly all models.

To intuitively understand this behavior, we compare the meta-train vocabulary with the meta-test vocabulary. On Amazon, 15095 of 28591 (52.8%) tokens in the meta-train vocabulary are not present in meta-test, and 21049 of 34545 (60.9%) tokens in the meta-test vocabulary are not present in meta-train. On Reuters, 3604 of 6372 (56.6%) meta-train tokens are not in meta-test, while 2481 of 5249 (47.2%) meta-test tokens are not in meta-test. Due to this lexical mismatch, finetuning will destroy the original geometry of the pretrained word embeddings, leading to poor generalization on unseen tasks.

| Method | | 20 News | | Amazon | | HuffPost | | RCV1 | | Reuters | | FewRel | | Average | |
|--------|--------|------|------|------|------|------|------|------|------|------|------|------|------|------|------|
| Rep. | Alg. | 1 shot | 5 shot | 1 shot | 5 shot | 1 shot | 5 shot | 1 shot | 5 shot | 1 shot | 5 shot | 1 shot | 5 shot | 1 shot | 5 shot |
| AVG | RR | 37.6 | 57.2 | 50.2 | 72.7 | 36.3 | 54.8 | 48.1 | 72.6 | 63.4 | 90.0 | 53.2 | 72.2 | 48.1 | 69.9 |
| IDF | RR | 44.8 | 64.3 | 60.2 | 79.7 | 37.6 | 59.5 | 48.6 | 72.8 | 69.1 | 93.0 | 55.6 | 75.3 | 52.6 | 74.1 |
| CNN | RR | 32.2 | 44.3 | 37.3 | 53.8 | 37.3 | 49.9 | 41.8 | 59.4 | 71.4 | 87.9 | 56.8 | 71.8 | 46.1 | 61.2 |
| OUR | RR | **52.1** | **68.3** | **62.6** | **81.1** | **43.0** | **63.5** | 54.1 | **75.3** | **81.8** | **96.0** | **67.1** | **83.5** | **60.1** | **78.0** |
| AVG | LR | 37.7 | 56.8 | 50.0 | 71.6 | 36.3 | 56.0 | 47.9 | 72.5 | 62.7 | 89.3 | 52.4 | 72.9 | 47.8 | 69.9 |
| IDF | LR | 46.1 | 64.2 | 60.5 | 79.9 | 38.7 | 59.3 | 47.3 | 72.5 | 68.1 | 93.0 | 55.5 | 74.9 | 52.7 | 74.0 |
| CNN | LR | 32.6 | 47.5 | 41.0 | 58.5 | 37.6 | 50.9 | 44.2 | 63.6 | 75.0 | 90.3 | 58.7 | 72.1 | 48.2 | 63.8 |
| OUR | LR | 49.7 | 67.6 | 61.6 | 80.7 | 41.9 | 61.9 | **54.4** | 74.1 | 79.5 | 95.6 | 65.8 | 82.8 | 58.8 | 77.1 |

Table 5: Results of 5-way 1-shot and 5-way 5-shot classification on six datasets using ridge regression vs. logistic regression with Newton's method.

| Method | | 5-way 5-shot Reuters | | 5-way 5-shot Amazon | |
|--------|--------|------|------|------|------|
| Rep. | Alg. | Fix embedding | Finetune embedding | Fix embedding | Finetune embedding |
| AVG | PROTO | $68.15_{\pm1.53}$ | $58.40_{\pm2.84}$ | $51.99_{\pm2.72}$ | $41.34_{\pm1.61}$ |
| IDF | PROTO | $72.07_{\pm2.69}$ | $65.93_{\pm3.08}$ | $59.24_{\pm1.10}$ | $48.37_{\pm2.41}$ |
| CNN | PROTO | $74.33_{\pm1.86}$ | $74.20_{\pm2.85}$ | $44.49_{\pm1.53}$ | $44.08_{\pm2.21}$ |
| AVG | MAML | $62.46_{\pm0.58}$ | $50.59_{\pm1.88}$ | $47.22_{\pm1.99}$ | $30.95_{\pm1.08}$ |
| IDF | MAML | $71.96_{\pm1.48}$ | $68.76_{\pm1.39}$ | $62.45_{\pm1.33}$ | $50.63_{\pm0.61}$ |
| CNN | MAML | $85.00_{\pm0.76}$ | $84.19_{\pm0.41}$ | $43.70_{\pm1.38}$ | $43.96_{\pm1.19}$ |
| AVG | RR | $90.00_{\pm0.49}$ | $86.87_{\pm0.60}$ | $72.78_{\pm0.23}$ | $75.65_{\pm0.51}$ |
| IDF | RR | $93.02_{\pm0.45}$ | $93.01_{\pm0.47}$ | $79.78_{\pm0.28}$ | $78.73_{\pm0.59}$ |
| CNN | RR | $87.93_{\pm1.49}$ | $85.47_{\pm0.57}$ | $53.89_{\pm1.54}$ | $52.63_{\pm2.02}$ |
| OUR | RR | $96.00_{\pm0.27}$ | $95.98_{\pm0.39}$ | $81.16_{\pm0.31}$ | $78.94_{\pm0.56}$ |

Table 6: Fixed vs. finetuned word embeddings during meta-training. Since meta-train and meta-test classes exhibit different word distributions, fixing the word embedding yields better generalization. That is, we avoid overfitting to the meta-train vocabulary and destroying the geometry of the pretrained embeddings.

## A.9 DISTRIBUTIONAL SIGNATURES WITH OTHER CLASSIFIERS

In this paper, we demonstrate the advantage of learning meta-knowledge on top of distributional signatures. We focus on a ridge regressor (Bertinetto et al., 2019) as our downstream classifier due to its simplicity and effectiveness. However, the idea of learning with distributional signatures is not limited to ridge regression; rather, we can combine distributional signatures with other learning methods such as prototypical networks (Snell et al., 2017) and induction networks (Geng et al., 2019).

**Prototypical networks** To augment a prototypical network with features learned from distributional signatures, we can construct per-class prototypes based on the attention-weighted representation $\phi(x)$ (Section 4.2). From Table 7, we see that learning with distributional signatures improves 5-way 1-shot accuracy by 9.9% and 5-way 5-shot accuracy by 16.7%, against the best prototypical network baseline for each dataset.

**Induction networks** We can also augment induction networks with meta-knowledge learned from distributional signatures. Specifically, we directly feed our attention-weighted representation $\phi(x)$ (Section 4.2) to the induction module. To avoid over-fitting on meta-train features, we replace the meta-learned relation module (see Geng et al. (2019) for details) by a parameter-free nearest neighbour predictor, similar to that of prototypical networks. From Table 7, we again see that learning with distributional signatures improves 5-way 1-shot accuracy by 14.3% and 5-way 5-shot accuracy by 25.1% on average.

These empirical results clearly demonstrate the benefit of our approach. However, both prototypical networks and induction networks perform consistently worse than the ridge regressor across all

| Method | | 20 News | | Amazon | | HuffPost | | RCV1 | | Reuters | | FewRel | | Average | |
|---|---|---|---|---|---|---|---|---|---|---|---|---|---|---|---|---|
| Rep. | Alg. | 1 shot | 5 shot | 1 shot | 5 shot | 1 shot | 5 shot | 1 shot | 5 shot | 1 shot | 5 shot | 1 shot | 5 shot | 1 shot | 5 shot |
| AVG | PROTO | 36.2 | 45.4 | 37.2 | 51.9 | 35.6 | 41.6 | 28.4 | 31.2 | 59.5 | 68.1 | 44.0 | 46.5 | 40.1 | 47.4 |
| IDF | PROTO | 37.8 | 46.5 | 41.9 | 59.2 | 34.8 | 50.2 | 32.1 | 35.6 | 61.0 | 72.1 | 43.0 | 61.9 | 41.8 | 54.2 |
| CNN | PROTO | 29.6 | 35.0 | 34.0 | 44.4 | 33.4 | 44.2 | 28.4 | 29.3 | 65.2 | 74.3 | 49.7 | 65.1 | 40.1 | 48.7 |
| OUR | PROTO | **42.4** | **59.5** | **52.6** | **73.9** | **40.6** | **58.9** | **45.1** | **63.6** | **72.2** | **93.9** | **57.5** | **75.3** | **51.7** | **70.9** |
| LSTM | INDUCT | 27.6 | 32.1 | 30.6 | 37.1 | 34.9 | 44.0 | 32.3 | 37.3 | 58.3 | 66.9 | 50.4 | 56.1 | 39.0 | 45.6 |
| OUR | INDUCT | **45.4** | **60.0** | **56.6** | **73.5** | **40.4** | **60.2** | **42.3** | **60.3** | **74.5** | **93.1** | **60.5** | **77.3** | **53.3** | **70.7** |

Table 7: Performance of prototypical networks and induction networks learned on lexical information vs. distributional signatures (OUR). LSTM with INDUCT is the original induction network architecture.

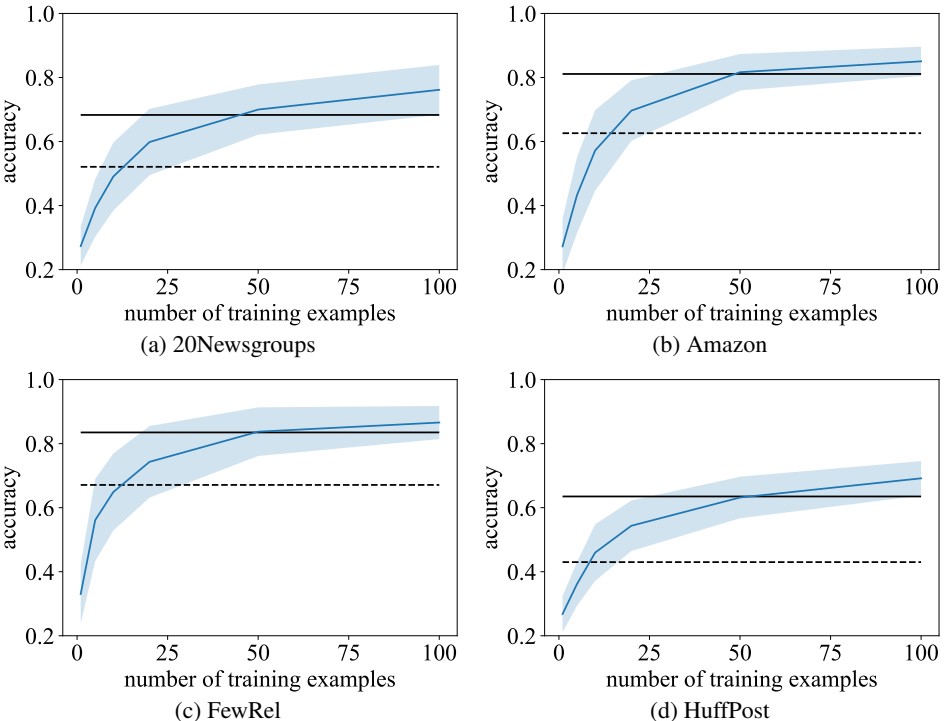

Figure 15: Learning curve for fully-supervised CNN classifier vs OUR. Blue indicates CNN accuracy, with standard deviation shaded. Solid horizontal line is OUR's 5-shot accuracy; dashed line is 1-shot accuracy. OUR's 5-shot performance is competitive, especially when the total number of labeled examples is small.

experiments. Our hypothesis is that the ridge regressor enables task-specific finetuning based on the support set, while the prototypical network and induction network directly compute nearest neighbours from the meta-learned metric space.

## A.10    FEW-SHOT LEARNING V.S. SUPERVISED LEARNING

To understand the extent of few-shot learning's utility, we compare our results with that of a fully-supervised CNN classifier, as we increase the number of training points. From Figure 15, we see that our model's 5-shot accuracy is approximately equivalent to a fully-supervised classifier trained on 50 examples.

## A.11 Implementation Details

We detail the implementations of our baselines here. All baseline code are available in our repository.

**CNN** For 1D convolution, we use filter windows of 3, 4, 5 with 50 feature maps each. We applied ReLU after max-over-time pooling.

**Prototypical Network** Prototypical network meta-learns a multi-layer perceptron to transform the input representation into an embedding space that is suitable for few-shot classification. If the input representation is learnable (e.g., CNN), the parameters for the input representation are also updated using meta-training. In the experiments, we use a MLP with one hidden layer and ReLU activation. The dimensions of both the hidden layer and the output layer are 300. We apply dropout with rate 0.1 to the hidden layer.

**MAML** MAML meta-learns an initialization such that the model can quickly adapt to new tasks after a few gradient steps. For prediction on the input representation, we use a MLP with one hidden layer of 300 ReLU units. We apply dropout with rate 0.1 to the hidden layer. During the MAML inner loop (adaptation stage), we perform ten updates with step size 0.1 (we empirically found this outperform one-step MAML). We backpropagate higher order gradients thoughout meta-training. During the MAML outerloop, we average the gradient across ten sampled tasks and use Adam with learning rate $10^3$ to update the parameters.

**Finetune** Chen et al. (2019) recently showed that fine-tuning a properly pre-trained classifier can achieve competitive performance when compared with the state-of-the-art meta-learning. Following their work, we explicitly reduce the intra-class variation during the pre-training stage. Similar to MAML, we use a MLP with one hidden layer (300 ReLU units) to make predictions from the input representation (e.g., CNN). During finetuning stage, we re-train the MLP from scratch and fine-tune the learnable parameters of the input representation. We stop fine-tuning once the gradient norm is less than $10^{-3}$.

**Induction network** Induction network consists of three modules: encoder module, induction module and relation module. The encoder module uses a biLSTM with self-attention to obtain a fix-length representation of each input example. The induction module performs dynamic routing to compute the class-specific prototype. The relation module uses a neural tensor layer to predict the relation association between each query example and the prototypes. Following Geng et al. (2019), we set the hidden state size of biLSTM to 256 (128 for each direction) and the attention dimension to 64. The iteration number in dynamic routing is set to 3. The dimension of the neural tensor layer is set to 100.

**P-MAML** P-MAML (Zhang et al., 2019) has two phases: masked language model pretraining (Devlin et al., 2018) and MAML (Finn et al., 2017). For pretraining, we used Hugging Face's language model finetuning code with default hyperparameters and BERT's pretrained base-uncased model. We applied early stopping when the validation perplexity failed to decrease for 2 epochs. After pretraining, we add a softmax layer on top of the representation of the [CLS] token (Devlin et al., 2018) to make predictions. During the MAML inner loop (adaptation stage), we perform ten updates with step size $10^{-3}$. Following Zhang et al. (2019), we do not consider higher order gradients. During the MAML outerloop, we average the gradient across ten sampled tasks and use Adam with learning rate $10^{-5}$ to update the parameters.

## A.12 RESULTS

This section contains experimental results with standard deviations.

| Rep. | Alg. | 20 News | Amazon | HuffPost | RCV1 | Reuters | FewRel |
|---|---|---|---|---|---|---|---|
| AVG | NN | 33.95 $\pm0.33$ | 46.76 $\pm0.14$ | 31.45 $\pm0.18$ | 43.76 $\pm0.14$ | 56.48 $\pm0.91$ | 47.58 $\pm0.38$ |
| IDF | NN | 38.88 $\pm0.33$ | 51.43 $\pm0.15$ | 31.53 $\pm0.18$ | 41.96 $\pm0.22$ | 57.76 $\pm0.91$ | 46.84 $\pm0.31$ |
| CNN | FT | 33.00 $\pm0.74$ | 45.71 $\pm0.86$ | 32.45 $\pm0.54$ | 40.33 $\pm1.47$ | 70.89 $\pm4.30$ | 54.08 $\pm0.33$ |
| AVG | PROTO | 36.25 $\pm0.33$ | 37.26 $\pm1.85$ | 35.68 $\pm1.25$ | 28.48 $\pm0.96$ | 59.54 $\pm1.48$ | 44.04 $\pm0.80$ |
| IDF | PROTO | 37.86 $\pm1.13$ | 41.91 $\pm1.17$ | 34.88 $\pm0.73$ | 32.14 $\pm0.51$ | 61.00 $\pm1.23$ | 43.09 $\pm1.15$ |
| CNN | PROTO | 29.67 $\pm1.02$ | 34.02 $\pm1.48$ | 33.49 $\pm0.79$ | 28.43 $\pm0.68$ | 65.22 $\pm1.52$ | 49.78 $\pm0.22$ |
| AVG | MAML | 33.74 $\pm0.32$ | 39.35 $\pm1.38$ | 36.14 $\pm1.23$ | 39.98 $\pm1.83$ | 54.55 $\pm1.08$ | 43.83 $\pm2.04$ |
| IDF | MAML | 37.26 $\pm1.62$ | 43.63 $\pm2.42$ | 38.95 $\pm0.48$ | 42.58 $\pm0.77$ | 61.46 $\pm1.50$ | 48.22 $\pm1.20$ |
| CNN | MAML | 28.98 $\pm1.62$ | 35.30 $\pm1.04$ | 34.12 $\pm0.86$ | 39.03 $\pm0.97$ | 66.62 $\pm1.89$ | 51.73 $\pm3.98$ |
| AVG | RR | 37.60 $\pm0.10$ | 50.25 $\pm0.23$ | 36.33 $\pm0.36$ | 48.17 $\pm0.17$ | 63.39 $\pm0.95$ | 53.25 $\pm1.01$ |
| IDF | RR | 44.83 $\pm1.07$ | 60.27 $\pm1.33$ | 37.68 $\pm0.96$ | 48.65 $\pm0.57$ | 69.12 $\pm1.87$ | 55.65 $\pm1.08$ |
| CNN | RR | 32.25 $\pm1.62$ | 37.30 $\pm0.80$ | 37.32 $\pm1.15$ | 41.81 $\pm1.49$ | 71.40 $\pm1.63$ | 56.83 $\pm2.30$ |
| OUR | | **52.17** $\pm0.65$ | **62.66** $\pm0.67$ | **43.09** $\pm0.16$ | **54.15** $\pm1.06$ | **81.81** $\pm1.61$ | **67.10** $\pm0.93$ |
| OUR w/o t($\cdot$) | | 50.15 $\pm1.62$ | 61.77 $\pm0.73$ | 42.09 $\pm0.37$ | 51.51 $\pm0.75$ | 76.71 $\pm1.44$ | 66.93 $\pm0.46$ |
| OUR w/o s($\cdot$) | | 41.99 $\pm0.69$ | 51.12 $\pm0.88$ | 40.16 $\pm0.23$ | 48.59 $\pm0.62$ | 78.15 $\pm0.99$ | 65.83 $\pm0.33$ |
| OUR w/o biLSTM | | 50.35 $\pm0.73$ | 61.95 $\pm0.40$ | 42.22 $\pm0.66$ | 51.88 $\pm0.84$ | 77.17 $\pm1.53$ | 66.42 $\pm0.35$ |
| OUR w EBD | | 39.68 $\pm2.60$ | 56.48 $\pm2.63$ | 40.64 $\pm0.84$ | 48.64 $\pm0.69$ | 81.68 $\pm2.21$ | 61.47 $\pm1.89$ |

Table 8: 5-way 1-shot classification. The bottom four rows present our ablation study.

| Rep. | Alg. | 20 News | Amazon | HuffPost | RCV1 | Reuters | FewRel |
|---|---|---|---|---|---|---|---|
| AVG | NN | 45.87 $\pm0.39$ | 60.36 $\pm0.20$ | 41.55 $\pm0.23$ | 60.84 $\pm0.19$ | 80.51 $\pm0.66$ | 60.67 $\pm0.30$ |
| IDF | NN | 51.94 $\pm0.20$ | 67.15 $\pm0.21$ | 42.35 $\pm0.15$ | 58.27 $\pm0.23$ | 82.88 $\pm0.57$ | 60.62 $\pm0.41$ |
| CNN | FT | 47.17 $\pm0.98$ | 63.91 $\pm1.20$ | 44.13 $\pm0.70$ | 62.34 $\pm0.56$ | 90.95 $\pm3.59$ | 71.19 $\pm0.57$ |
| AVG | PROTO | 45.42 $\pm1.36$ | 51.99 $\pm2.72$ | 41.67 $\pm1.13$ | 31.22 $\pm1.53$ | 68.15 $\pm1.53$ | 46.55 $\pm1.58$ |
| IDF | PROTO | 46.53 $\pm1.44$ | 59.24 $\pm1.10$ | 50.24 $\pm0.94$ | 35.63 $\pm0.83$ | 72.07 $\pm2.69$ | 61.99 $\pm1.82$ |
| CNN | PROTO | 35.09 $\pm0.71$ | 44.49 $\pm1.53$ | 44.21 $\pm0.58$ | 29.33 $\pm0.81$ | 74.33 $\pm1.86$ | 65.16 $\pm0.99$ |
| AVG | MAML | 43.92 $\pm1.07$ | 47.22 $\pm1.99$ | 49.69 $\pm0.54$ | 50.69 $\pm1.03$ | 62.46 $\pm0.58$ | 57.87 $\pm1.86$ |
| IDF | MAML | 48.62 $\pm1.31$ | 62.45 $\pm1.33$ | 53.70 $\pm0.29$ | 54.14 $\pm0.72$ | 71.96 $\pm1.48$ | 65.80 $\pm1.19$ |
| CNN | MAML | 36.79 $\pm0.78$ | 43.70 $\pm1.38$ | 45.89 $\pm0.53$ | 51.15 $\pm0.39$ | 85.00 $\pm0.76$ | 66.90 $\pm3.23$ |
| AVG | RR | 57.24 $\pm0.19$ | 72.78 $\pm0.23$ | 54.86 $\pm0.21$ | 72.62 $\pm0.17$ | 90.00 $\pm0.49$ | 72.22 $\pm0.16$ |
| IDF | RR | 64.35 $\pm0.54$ | 79.78 $\pm0.28$ | 59.56 $\pm1.78$ | 72.85 $\pm0.21$ | 93.02 $\pm0.45$ | 75.30 $\pm0.34$ |
| CNN | RR | 44.32 $\pm0.44$ | 53.89 $\pm1.54$ | 49.96 $\pm0.20$ | 59.47 $\pm0.58$ | 87.93 $\pm1.49$ | 71.81 $\pm1.25$ |
| OUR | | **68.33** $\pm0.17$ | **81.16** $\pm0.31$ | **63.51** $\pm0.10$ | **75.38** $\pm1.12$ | **96.00** $\pm0.27$ | **83.53** $\pm0.27$ |
| OUR w/o t($\cdot$) | | 67.59 $\pm0.57$ | 80.58 $\pm0.19$ | 60.86 $\pm0.19$ | 75.12 $\pm0.90$ | 93.71 $\pm0.84$ | 83.20 $\pm0.39$ |
| OUR w/o s($\cdot$) | | 60.77 $\pm0.52$ | 75.37 $\pm0.27$ | 60.29 $\pm0.35$ | 72.84 $\pm0.23$ | 94.79 $\pm0.32$ | 82.62 $\pm0.20$ |
| OUR w/o biLSTM | | 66.99 $\pm0.32$ | 80.90 $\pm0.36$ | 63.04 $\pm0.20$ | 74.19 $\pm0.22$ | 95.36 $\pm0.39$ | 82.90 $\pm0.21$ |
| OUR w EBD | | 57.52 $\pm2.39$ | 76.34 $\pm0.65$ | 58.58 $\pm1.09$ | 71.50 $\pm0.28$ | 95.76 $\pm0.53$ | 80.88 $\pm1.18$ |

Table 9: 5-way 5-shot classification. The bottom four rows present our ablation study.

| talk.politics.mideast | sci.space | misc.forsale | talk.politics.misc | comp.graphics |
|---|---|---|---|---|
| israel | space | sale | president | image |
| armenian | nasa | 00 | cramer | graphics |
| turkish | launch | shipping | mr | jpeg |
| armenians | orbit | offer | stephanopoulos | images |
| israeli | shuttle | condition | people | gif |
| jews | moon | 1st | government | format |
| armenia | henry | price | optilink | file |
| arab | earth | forsale | myers | 3d |
| people | mission | asking | clayton | ftp |
| jewish | solar | comics | gay | color |

| sci.crypt | comp.windows.x | comp.os.ms-windows.misc | talk.politics.guns | talk.religion.misc |
|---|---|---|---|---|
| key | window | ax | gun | god |
| clipper | xx | max | guns | jesus |
| encryption | motif | windows | fbi | sandvik |
| chip | server | g9v | firearms | christian |
| keys | widget | b8f | atf | bible |
| security | file | a86 | batf | jehovah |
| privacy | xterm | 145 | weapons | christ |
| government | x11 | pl | people | lord |
| escrow | entry | 1d9 | waco | kent |
| des | dos | 34u | cdt | brian |

| rec.autos | sci.med | comp.sys.mac.hardware | sci.electronics | rec.sport.hockey |
|---|---|---|---|---|
| car | medical | mac | circuit | hockey |
| cars | disease | apple | wire | game |
| engine | msg | centris | ground | team |
| ford | cancer | quadra | wiring | nhl |
| oil | health | lc | voltage | play |
| dealer | patients | monitor | battery | season |
| callison | doctor | duo | copy | games |
| mustang | hiv | nubus | amp | 25 |
| com | food | drive | electronics | ca |
| autos | diet | simms | audio | pit |

| alt.atheism | rec.motorcycles | comp.sys.ibm.pc.hardware | rec.sport.baseball | soc.religion.christian |
|---|---|---|---|---|
| god | bike | scsi | baseball | god |
| atheism | dod | drive | game | church |
| atheists | ride | ide | year | jesus |
| keith | bmw | controller | team | christ |
| livesey | com | card | players | sin |
| morality | riding | bus | games | christians |
| religion | bikes | drives | hit | christian |
| moral | motorcycle | bios | braves | rutgers |
| islamic | dog | disk | runs | bible |
| say | rider | pc | pitcher | faith |

Table 10: Top 10 LMI-ranked words for each class in 20 Newsgroup. Class names are shown in italic. Different class exhibit different salient features.

