# OpenReview forum: "Few-shot Text Classification with Distributional Signatures"
_ICLR.cc/2020/Conference — Accept (Poster)_

### Official Review · AnonReviewer1 · 2019-10-23
**Official Blind Review #1**

**Rating:** 3

**Review:**

This paper studies the effects of using function of ngram statistics as feature to generate attention score per word. The attention score is then used as weights to aggregate document embedding by doing a weighted average on word embedding. The output is finally fed into a ridge regressor to do the final predictions on target labels.

Main comments:
This paper has a clear motivation and decent experimental results (though some concern on baseline models, see below). The introduction of using distributional signature to derive attention scores seems interesting and a novel contribution. However I was not able to fully understand the intuition behind the benefit of doing attention mechanism on top of ngram statistics (see my question below as well).
Also the reference/baseline models used in the experiment might not be strong enough. If you could compare your model with some latest algorithms proposed in the few-shot-learning communities, that would be more convincing as well.
To list a few:
* P-MAML: [Zhang et al., 2019]
* Induction-Network-Routing: [Geng et al., 2019]
* ROBUSTTC-FSL [Yu et al., 2018]

I am leaning to give a "weak reject" based on my current knowledge and understanding of the paper. But I will be willing to revisit the decision after we get feedback from the author(s).

In particular, I would be glad if the author could clarify the questions below.

* From table 1, it seems Method IDF+RR is a competitive model. IIUC, the statistics of s(.) is highly correlated with IDF which also indicates general word importance in corpus. My questions are that,
1) regarding ablation test "OUR w/o biLSTM", how is $h$ calculated in this case (without biLSTM)?
2) since each word is represented based on two statistical number (map function by t(.) and s(.)), can you give any intuitive explanation that why getting attention score from that makes sense?
3) do you have any experiments using the distributional signature as a common feature in standard text classification problems? In other words, is this method only (significantly) beneficial to few-short-learning? If it is also useful in general text classification task, it would be a good "plus" here.

* From table 2, can you explain why CNN+RR benefits a lot from the BERT embedding? Actually it gets more percentage of improvement than the model "OUR".

* For all the usages of pre-trained embedding (fasttext or BERT), are you further finetuning the embedding parameters during your training? Or you freeze the embedding parameters?

[Zhang et al., 2019] Ningyu Zhang et al., Improving Few-shot Text Classification via Pretrained Language Representations. arXiv preprint arXiv: 1908.08788
[Geng et al., 2019] Ruiying Geng, Binhua Li, Yongbin Li, Yuxiao Ye, Ping Jian, and Jian Sun. 2019. Few-shot text classification with induction network. arXiv preprint arXiv:1902.10482.
 [Yu et al., 2018] Mo Yu, Xiaoxiao Guo, Jinfeng Yi, Shiyu Chang, Saloni Potdar, Yu Cheng, Gerald Tesauro, Haoyu Wang,
and Bowen Zhou. 2018. Diverse few-shot text classification with multiple metrics. arXiv preprint arXiv:1805.07513

**Experience Assessment:**

I have read many papers in this area.

**Review Assessment: Checking Correctness Of Derivations And Theory:**

I assessed the sensibility of the derivations and theory.

**Review Assessment: Checking Correctness Of Experiments:**

I assessed the sensibility of the experiments.

**Review Assessment: Thoroughness In Paper Reading:**

I read the paper at least twice and used my best judgement in assessing the paper.

---

> ### Author Response · Authors · 2019-11-09
> **Intuition of distributional signatures and additional experiments**
>
> Thank you for the detailed comments and suggestions!
>
> The main idea of the paper is that if we want to learn transferable knowledge, methods that memorize word identities will fail when the word distribution shifts. By learning meta-knowledge on top of n-gram statistics, class-specific words will still be important (and common stop words unimportant), even if the actual words themselves change. (More specific example regarding our two statistics below.)
>
> Additional Experiments: Based on the suggestions, we have compared our work to P-MAML and Induction Network Routing. Detailed results are located in Appendix A.5.
>
> On average, our method outperforms P-MAML by 18.5% on 1-shot and 19.3% on 5-shot, and Induction Networks by 21.1% on 1-shot and 32.4% on 5-shot (Table 4). While both P-MAML and Induction Networks are able to overfit the meta-train data easily, they are unable to generalize when faced with lexical mismatch (Figure 9).
>
> Furthermore, we show that we can improve Induction Networks by replacing its lexically-aware encoder with our attention-weighted representation learned from distributional signatures  (Appendix A.9). On average, distributional signatures increase Induction Networks accuracy by 14.3% on 1-shot and 25.1% on 5-shot.
>
> RobustTC-FSL is not directly applicable to our setting since it considers a fixed set of tasks during meta-training (e.g. binary sentiment classification across 23 Amazon domains) and utilizes their cross-task transferability.
>
> Questions:
> * s(.) and IDF both indicate general word importance. We experimented with both during our development stage and found that they perform similarly when used in our attention generator (idf 77.8 vs s(.) 78.0 for 5 shot classifications averaged across 6 datasets). We choose the current formulation as it is more interpretable in context of robustness against word-substitution perturbation.
>
> 1) We apply an MLP on top of the [s(); t()] at each position, where ; denotes concatenation. After that we applied softmax over the output of the MLP. This MLP has 2 inputs, 50 hidden units (ReLU activation) and 1 output.
>
> 2) One indicates word importance for general classification (estimated from source pool) and the other indicates how important the feature is for this particular task (a rough estimate).
>
> For example, suppose we have lots of data from political and sports news, and we want to expand into arts news. General word importance (learned from politics and sports) can tell us that words like “the” and “we” are not useful, so we learn to ignore them. However, politics and sports news also have no use for arts-specific words, like “painting” or “performance.” Thus, we require task-specific word importance (learned from few arts examples) to refine our understanding of useful words.
>
> 3) The general idea of “distributional signatures” is not new. Prior to the age of deep learning, linear SVM + TF-IDF was considered a strong baseline to beat, and more recently, Arora 2016 showed that SIF-weighted representations (statistics used for s(.) in our model) do outperform LSTMs/CNNs on some (standard) tasks. In our setting, we noted that the idea may also be helpful for few-shot classification, as these statistics are more transferable across tasks. For general classification tasks with lots of annotation, the representation power of distribution signatures may be limited (though this is slightly beyond the scope of our paper).
>
> * BERT is contextual, so the embedding of one word represents not only itself, but also its surroundings. Correspondingly, if a CNN downweights an important word from an unseen class, its adjacent words still contain information about that word. This means that it is less “costly” to ignore important words from unseen classes, as a result of overfitting on seen classes. For OUR, this means that we don’t have to be as precise about picking out each important word.
>
> * Since the vocabulary of meta-train classes and meta-test classes may be very different, we freeze the pre-trained embeddings (Fasttext or BERT) during meta training. This is to avoid disrupting the inherent geometry of word embeddings, as finetuning will cause these embeddings to lose the relationship between meta-train vocabulary (seen during finetuning) and meta-test vocabulary (not seen, and thus not optimized for). Empirically, we show that freezing word embeddings outperforms finetuning (Table 6).
>
> We hope we have adequately addressed your concerns. Please let us know if you have any more questions. Thank you!

---

### Official Review · AnonReviewer2 · 2019-10-24
**Official Blind Review #2**

**Rating:** 1

**Review:**

This paper focuses on applying meta learning approaches to text classification.

The primary contribution is an attention mechanism based on word statistics --- most importantly word frequency. This novel attention mechanism is motivated by the observation that the base units in text (lexemes) are more likely to have task specific interpretations than lower level patterns in vision. And, while a lexeme based attention mechanism trained on one task may not transfer well to other tasks, a mechanism based on coarser word statistics is less likely to focus in on task specific patterns.

A secondary contribution is the use of ridge regression [1] to perform meta-learning for text classification.

The paper presents experiments on a number of text classification tasks from the NLP literature. Aside from the new attention mechanism and the use of ridge regression, the proposed approach makes use of FastText word embeddings or BERT sentence representations, depending on the task. The paper demonstrates significant improvements over baselines that use other methods of aggregating word representations. All of baselines were implemented for this paper.

The idea of using coarse statistical signatures to calculate attention is an interesting one. However, I have concerns about both the clarity of this paper and the lack of clear comparison to previous work.

== Clarity ==

Many of the details of the model and learning approach are vaguely discussed, or relegated to Figures 4 & 5. I think the paper would benefit from a more formal definition of the entire learning procedure.

== Comparison to previous work ==

This paper seems to be following the standard FewRel experimental setup. Also, the RCV1 experiments seem to follow the [2] which was cited by the in the paper under review. However, it is not clear if the setups are the same or if the numbers are comparable.

I am not sure about the existence of comparable results for the other tasks, but for FewRel at least the baselines presented here significantly underperform other papers' reports of equivalent models.

  - The paper from [3] that introduced FewRel reported 69.2 / 84.8 for CNN based prototypical networks --- far above the 49.8 / 65.2 reported here.

  - [4] found that a BERT model with no FewRel specific training at all achieves 72.9% on the 5way/1shot task. Which is above all of the BERT based models reported in Table 2.

I may be missing something, but if these numbers are actually not comparable then this paper should contain an explanation of how the experimental setup differs. And if the setup is actually the same as previous work, I expect to see a comparison of results.

[1] https://openreview.net/pdf?id=HyxnZh0ct7
[2] https://openreview.net/forum?id=SyxMWh09KX
[3] https://www.aclweb.org/anthology/D18-1514/
[4] https://arxiv.org/abs/1906.03158

**Experience Assessment:**

I have published in this field for several years.

**Review Assessment: Checking Correctness Of Derivations And Theory:**

I assessed the sensibility of the derivations and theory.

**Review Assessment: Checking Correctness Of Experiments:**

I assessed the sensibility of the experiments.

**Review Assessment: Thoroughness In Paper Reading:**

I read the paper at least twice and used my best judgement in assessing the paper.

---

> ### Author Response · Authors · 2019-11-07
> **Clarification regarding our model and experiments**
>
> [This comment has been updated to reflect changes in paper numbering]
>
> Thank you for the detailed comments!
>
> Clarity: Figures 4 and 5 are provided as summaries of the learning procedure and model; all components are formally defined in Sections 3 and 4, with implementation details in Section 5.
>
> Model: All notation is formally defined within the main text, with additional implementation details in the Appendix.
> - For the attention generator, the raw statistics s(.) and t(.) are defined in equations 1 and 2 respectively. Implementation details regarding t(.) are found in Appendix A.1, as cited. The attention score \alpha is defined in equation 3. Details for the biLSTM are noted in Section 5.
> - For the ridge regressor, the weighted representations \phi are defined in equation 4, and W is written out in equation 6.
>
> Learning Procedure: Based on the suggestions, we have added Appendix A.3 containing pseudocode for our entire learning procedure. Hyperparameters for training are already noted in Section 5.
>
> Experimental Setup: This paper does not follow the standard FewRel setup, as noted by Appendix A.4 and Table 3 (referenced in Section 5.1). We combine the train/val data of FewRel (test is not publicly available) and split it further into train/val/test.
>
> For data splits, we consider two settings: “easy split” randomly permuted classes and divided into train/val/test, and “hard split” selected train/val/test classes based on class hierarchies or similar heuristics, such that train classes are distant from val/test. Following [a], we use the “hard split” to test the meta-learning algorithm’s generalization capacity when testing tasks may not come from the same domain as training tasks. Please see the Appendix for details regarding each dataset.
>
> For RCV1 and Reuters, [b] does not provide pruned classes, data splits, or code, so we cannot directly compare. Our code (including all baselines), data splits, and processed data are publicly available.
>
> We hope we have adequately addressed your concerns. Please let us know if you have any more questions. Thank you!
>
> References
>
> [a] Zero-Shot Learning - The Good, the Bad and the Ugly
> [b] Attentive Task-Agnostic Meta-Learning for Few-Shot Text Classification

---

### Official Review · AnonReviewer3 · 2019-10-26
**Official Blind Review #3**

**Rating:** 6

**Review:**

UPDATE: Based on the extensive improvements by the authors, I have updated my rating. However, I still have doubts about the potential of this approach to reach practically useful levels of accuracy.

This paper introduces a simple method to weight pretrained lexical features for use in meta learning of few-shot text classification. The method boils down to weighting word inut features, in the form of pretrained word-embeddings, by attention computed from inverse document frequency and class local mutual information. The idea is that this measure of feature informativeness transfers between tasks, whereas lexical features themselves are highly task-specific. The approach is well motivated and is empirically shown to outperform existing approaches to few-shot text classification with a significant margin.

While the improvement over existing approaches is quite substantial, I believe the paper should not be accepted to ICLR for the following reasons. First, the contribution is quite limited and not particularly novel. While two weight functions are proposed, the majority of improvement comes simply from normalizing IDF with attention. Based on existing work on delexicalized features for NLP tasks such as parsing, this is quite a straightforward extension. Given the limited contribution, a short paper seems a better fit. Second, the approach simply trades variance for bias. This brings us to the question of how likely the approach is to be a building block in bringing us towards a pratically useful few-shot classification method. Given the weak representational power of the model, I believe this is unlikely. I see a situation similar to syntactic parsing for low-resource languages, where a collection of simple techniques similar in spirit to the current approach, like delexicalization, brought results far above the naive baselines, but never approached practically useful results. I think this is a crucial point to address in meta-learning research in general to make sure we’re not just solving a toy problem with tailored heuristics.

Additional notes:

What is the motivation for using a BiLSTM to combine inverse document frequency and inverse class entropy? Is the sequence information at all useful, or would a simple projection and nonlinearity give the same result?

The theoretical analysis is completely self-evident from the definition of the feature space. Replaing a feature with an equivalent feature of course gives the same result and I don’t see the need to “mathematize” this.

The effect of approximating logistic regression with linear regression + calibration is not analyzed and it is not clear what the effect of this approximation is in the text classification scenario. I would suggest to compare to differentiating through a direct optimization of the logistic formulation, for example with Newton’s method, or plain SGD, as in Bertinetto et al. (2019).

Table 1. Why not run the attention-based feature aggregator together with all algorithms. The main contribution is at the input representation level, and this should be applicable across algorithms. In fact, if we remove the BiLSTM which seems to have a very small effect the representation function does not contain learnable parameters in itself.

Please provide the average across datasets in Table 1.

**Experience Assessment:**

I have published in this field for several years.

**Review Assessment: Checking Correctness Of Derivations And Theory:**

I carefully checked the derivations and theory.

**Review Assessment: Checking Correctness Of Experiments:**

I carefully checked the experiments.

**Review Assessment: Thoroughness In Paper Reading:**

I read the paper thoroughly.

---

> ### Author Response · Authors · 2019-11-07
> **Contributions and additional notes**
>
> [This comment has been updated to reflect changes in paper numbering]
>
> Thank you for your detailed comments!
>
> We would like to emphasize that this paper makes unique and valuable contributions to the few-shot learning community.
>
> - It is the first to identify that meta-learning algorithms only memorize useful training features, instead of actually learning to adapt quickly to new settings. Thus, standard meta-learning algorithms may not generalize in NLP, when word distributions differ vastly among tasks. Later on, [a] reported similar findings for vision.
>
> - The main contribution of this paper is not a specific set of delexicalised features for an end task (classification/parsing), but rather a meta-learning approach towards generalizable representations in low-resource settings. In few-shot learning, we are the first to propose that meta-knowledge can be learned on top of feature statistics, instead of features themselves (which are not transferable).
>
> - We provide the largest publicly available few-shot text classification benchmarks to the community. Previous work [b, c] focus on binary sentiment classifiers across different domains, while we are interested in multi-class text classification. The setup of [d] is similar to ours, but neither their datasets nor code are publicly available, and they only run experiments on two datasets (RCV1, Reuters).
>
> In terms of raw representation power, our model is definitely weaker than fully-lexical models. Instead, the key benefit is that it generalizes in low-resource settings (Figure 6). Quantitatively, our 5-shot accuracy is approximately equivalent to a fully supervised model trained on 50 examples (Figure 14).
>
> While 50 examples does not seem like very much, even for low-resource languages, we would like to present one case we have faced where we cannot hope for this many. We are given a set of pathology reports, which detail the diagnoses of various tissues in a patient’s body (e.g. lung, ovary). We want to predict whether the patient has a disease. This is not a “hard” task, and a CNN would perform quite well, given enough data. However, these individual diagnoses correspond to potentially thousands of diseases. The total number of {tissue diagnosis, disease} pairs far exceeds the number of reports. Often, each pair only has one or two examples. In this case, it is unrealistic to train a fully-supervised classifier for each pair.
>
> If we resort to standard meta-learning techniques, learning lexical information from well-represented disease/tissue pairs may be distracting for rarer cases: there is a staggering number of medical terms regarding each disease/tissue, few of these terms overlap across tasks. On the other hand, if we work with distributional statistics, we can learn to ignore terms that are not useful, regardless of task. As a result, we can better focus on words that are meaningful for classification.
>
> Additional Notes:
> 1. We provide the comparison in the ablation study (Table 1, OUR w/o biLSTM), where we learn a projection from the two statistics using an MLP. On average, the biLSTM improves accuracy from 77.2 to 78.0 (best baseline RR+IDF is 74.1).
> 2. The idea was to provide some intuition about distributional signatures, if the reader finds them clearer to understand in this way.
> 3. We have added experiments comparing ridge regression vs. logistic regression with Newton’s method in Appendix A.7. On average, RR+OUR performs slightly better than LR+OUR (78.0 vs. 77.1 on 5-shot, 60.1 vs. 58.8 on 1-shot), but both significantly outperform the best baselines (RR+IDF, 74.1 on 5-shot; LR+IDF, 52.7 on 1-shot).
> 4. In this paper, we focused on RR as the main downstream predictor due to its simplicity and effectiveness. However, our experiments show that the attention-based feature aggregator also improves performance for other algorithms. Appendix A.9 contains results from distributional signatures + other classifiers. For prototypical networks, we improve 1-shot accuracy by 9.9 and 5-shot accuracy by 16.7 on average across 6 datasets, compared to the best baseline. For induction networks, we improve average accuracy by 14.3% on 1-shot and 25.1% on 5-shot (Table 7).
> 5. Done, thank you for the suggestion.
>
> We hope we have adequately addressed your concerns. Please let us know if you have any more questions. Thank you!
>
>
> References
> [a] Rapid Learning or Feature Reuse? Towards Understanding the Effectiveness of MAML
> [b] Diverse Few-Shot Text Classification with Multiple Metrics
> [c] Induction Networks for Few-Shot Text Classification
> [d] Attentive Task-Agnostic Meta-Learning for Few-Shot Text Classification

---

### Author Response · Authors · 2019-11-09
**To all reviewers**

Thank you again for your detailed comments and suggestions. We would like to summarize the main contributions of our paper, as well as our recent updates to the paper. In addition, our code base has been updated with the latest experiments.

We would like to emphasize that this paper makes unique and valuable contributions to the few-shot learning community.

1. It is the first to identify that meta-learning algorithms only memorize features that are useful during meta-training, instead of actually learning to adapt quickly to new settings.

2. The main contribution of this paper is not a specific set of delexicalised features for an end task (classification/parsing), but rather a meta-learning approach towards generalizable representations. Meta-knowledge learned on top of distributional signatures can be used towards any downstream classifier to improve performance in low-resource settings.

3. We provide the largest publicly available few-shot text classification benchmarks to the community. Both our code (including all reported baselines) and data splits are available for reproducibility.

Additional experiments:

1. Comparison to other baselines.
Based on suggestions, we have also compared our work to Induction Network and P-MAML. We observe that our method vastly outperforms these methods, as they both build meta-knowledge from lexical features which do not generalize to different word distributions. (Table 4, Appendix A.5).

2. Distributional signatures improve other classifiers too.
We show that we can improve Prototypical Networks and Induction Network Routing by replacing their input representations with our attention-weighted representation learned from distributional signatures (Table 7, Appendix A.9).

3. Restricting meta-knowledge to distributional signatures is crucial.
When we feed the attention generator word embeddings together with the distributional signatures, performance drops significantly, as the model can perform much better on meta-train by focusing on non-transferable lexical features (Table 1, last row).

4. Finetuning BERT/FastText disrupts word embedding geometry due to lexical mismatch.
Methods generally perform worse when word embeddings are finetuned (Table 6).

5. Effect of softmax calibration with ridge regression.
Replacing ridge regression with logistic regression + Newton’s method yields no significant difference in results (Table 5).

---

### Decision · Program_Chairs · 2019-12-19

**Decision:**

Accept (Poster)

**Comment:**

This paper proposes a meta-learning approach for few-shot text classification. The main idea is to use an attention mechanism over the distributional signatures of the inputs to weight word importance. Experiments on text classification datasets show that the proposed method improves over baselines in 1-shot and 5-shot settings.

The paper addresses an important problem of learning from a few labeled examples. The proposed approach makes sense and the results clearly show the strength of the proposed approach.

R1 had some questions regarding the proposed method and experimental details. I believe this have been addressed by the authors in their rebuttal.

R2 suggested that the authors clarified their experimental setup with respect to prior work and improved the clarity of their paper. The authors have made some adjustments based on this feedback, including adding new sections in the appendix.

R3 had concerns regarding the contribution of the approach and whether it trades variance for bias. The authors have addressed most of these concerns and R3 has updated their review accordingly.

I think all the reviewers gave valuable feedbacks that have been incorporated by the authors to improve their paper. While the overall scores remain low, I believe that they would have been increased had R1 and R2 reassessed the revised submission. I recommend to accept this paper.